# Effects of quality-based procedure hospital funding reform in Ontario, Canada: An interrupted time series study

**Alvin Ho-ting Li**[1,2]*, **Karen S. Palmer**[3,4], **Monica Taljaard**[1,5], **J. Michael Paterson**[2,6], **Adalsteinn Brown**[2,6,7,8], **Anjie Huang**[2], **Husayn Marani**[3,6], **Lauren Lapointe-Shaw**[9], **Daniel Pincus**[2,6,10], **Marian S. Wettstein**[2,6,11], **Girish S. Kulkarni**[2,11,12], **David Wasserstein**[10,13], **Noah Ivers**[2,3,14]

**1** Ottawa Hospital Research Institute, Ottawa, Ontario, Canada, **2** Institute for Clinical Evaluative Sciences, Toronto, Ontario, Canada, **3** Women's College Research Institute, Women's College Hospital, Toronto, Ontario, Canada, **4** Faculty of Health Sciences, Simon Fraser University, Burnaby, British Columbia, Canada, **5** School of Epidemiology and Public Health, University of Ottawa, Ottawa, Ontario, Canada, **6** Institute of Health Policy, Management and Evaluation, University of Toronto, Toronto, Ontario, Canada, **7** Dalla Lana School of Public Health, University of Toronto, Toronto, Ontario, Canada, **8** Department of Obstetrics and Gynecology, Faculty of Medicine, University of Toronto, Toronto, Ontario, Canada, **9** Department of Medicine, University of Toronto, Toronto, Ontario, Canada, **10** Division of Orthopaedic Surgery, Department of Surgery, University of Toronto, Toronto, Canada, **11** Department of Surgical Oncology (Urology), Princess Margaret Cancer Centre, University Health Network, University of Toronto, Toronto, Ontario, Canada, **12** Division of Urology, Department of Surgery, Faculty of Medicine, University of Toronto, Toronto, Ontario, Canada, **13** Sunnybrook Research Institute, Sunnybrook Health Sciences Centre, Toronto, Ontario, Canada, **14** Department of Family and Community Medicine and Institute of Health Policy, Management and Evaluation, University of Toronto, Toronto, Ontario, Canada

* ali66@uwo.ca

## Abstract

### Background

The Government of Ontario, Canada, announced hospital funding reforms in 2011, including Quality-based Procedures (QBPs) involving pre-set funds for managing patients with specific diagnoses/procedures. A key goal was to improve quality of care across the jurisdiction.

### Methods

Interrupted time series evaluated the policy change, focusing on four QBPs (congestive heart failure, hip fracture surgery, pneumonia, prostate cancer surgery), on patients hospitalized 2010–2017. Outcomes included return to hospital or death within 30 days, acute length of stay (LOS), volume of admissions, and patient characteristics.

### Results

At 2 years post-QBPs, the percentage of hip fracture patients who returned to hospital or died was 3.13% higher in absolute terms (95% CI: 0.37% to 5.89%) than if QBPs had not been introduced. There were no other statistically significant changes for return to hospital or death. For LOS, the only statistically significant change was an increase for prostate cancer surgery of 0.33 days (95% CI: 0.07 to 0.59). Volume increased for congestive heart

**Data Availability Statement:** The dataset from this study is held securely in coded form at ICES. While legal data sharing agreements between ICES and data providers (e.g., health organizations and

government) prohibit ICES from making the dataset publicly available, access may be granted to those who meet pre-specified criteria for confidential access, available at www.ices.on.ca/DAS (email: das@ices.on.ca). The full dataset creation plan and underlying analytic code are available as Supporting Information files, understanding that the computer programs may rely upon coding templates or macros that are unique to ICES and are therefore either inaccessible or may require modification.

**Funding:** This work was funded through an Ontario Strategy for Patient Oriented Research Support Unit (OSSU) Impact Award, which was, in turn, funded by the Canadian Institutes of Health Research and the Government of Ontario. The funders had no role in study design, data collection and analysis, decision to publish, or preparation of the manuscript. This study was also supported by ICES, which is funded by an annual grant from the Ontario Ministry of Health and Long-Term Care (MOHLTC). The opinions, results and conclusions reported in this paper are those of the authors and are independent from the funding sources. No endorsement by ICES or the Ontario MOHLTC is intended or should be inferred. Parts of this material are based on data and information compiled and provided by the Canadian Institute for Health Information (CIHI). However, the analyses, conclusions, opinions and statements expressed herein are those of the authors, and not necessarily those of CIHI.

**Competing interests:** The authors have read the journal's policy and have the following competing interests: NI, AB, KP, HM report funding from Ontario Strategy for Patient Oriented Research Support Unit (OSSU) Impact Award during the conduct of the study; OSSU was in turn funded by the Canadian Institutes of Health Research (CIHR) and the Government of Ontario. NI is supported by a Canada Research Chair (Tier 2) in Implementation of Evidence-Based Practice and a clinician scholar award from the Department of Family and Community Medicine, University of Toronto, unrelated to this work. AB is a former senior official in the Ontario Government and serves on the current Premier's Council on Health. AL reports support from a CIHR Fellowship Award. This does not alter our adherence to PLOS ONE policies on sharing data and materials.

failure admissions by 80 patients (95% CI: 2 to 159) and decreased for hip fracture surgery by 138 patients (95% CI: -183 to -93) but did not change for pneumonia or prostate cancer surgery. The percentage of patients who lived in the lowest neighborhood income quintile increased slightly for those diagnosed with congestive heart failure (1.89%; 95% CI: 0.51% to 3.27%) and decreased for those who underwent prostate cancer surgery (-2.08%; 95% CI: -3.74% to -0.43%).

## Interpretation

This policy initiative involving a change to hospital funding for certain conditions was not associated with substantial, jurisdictional-level changes in access or quality.

## Introduction

Policymakers worldwide are experimenting with hospital funding models to improve system performance [1–3]. Although such reforms may contribute to improvements in resource allocation and patient outcomes, they may also invoke unintended consequences [4–6].

In April 2011, the Government of Ontario, Canada, announced a multi-year phased-in implementation of "patient-based" hospital funding [7]. These hospital funding reforms reduced reliance on global hospital budgets (i.e., fixed annual amount based largely on historical spending) by introducing two new components to hospital funding: Health Based Allocation Model (HBAM), organizational-level funding based on service and patient characteristics; and Quality-Based Procedures (QBPs), a novel approach to hospital funding sharing some characteristics with activity-based funding (ABF) [8]. QBPs consist of pre-set reimbursement rates for managing patients with specific diagnoses or those undergoing specific procedures, coupled with best-practice clinical handbooks for each QBP [7]. Between April 2012-April 2016, 19 priority QBPs were implemented for a range of medical and surgical conditions [9].

The stated goal of QBPs was to "facilitate adoption of best clinical evidence-informed practices" and appropriately reduce "variation in costs and practice across the province while improving outcomes" [10]. The provincial government's rationale and assumed mechanism of action for QBPs was as follows: "QBPs are specific clusters of patient services that offer opportunities for health care providers to share best practices and will allow the system to provide even better quality care, while increasing system efficiencies. By promoting the adoption of clinical evidence-informed practices, clinical practice variation should be reduced across the province while improving patient outcomes to ensure that patients receive the right care, in the right place, at the right time" [11].

Like all "patient focused" activity-based funding systems, QBPs established a prospective payment rate based on service type and volume. Funding was carved out of hospitals' global budgets and then reallocated to hospitals at the start of the relevant fiscal year as a fixed fee and fixed volume, for each QBP procedure or diagnosis. The fixed volume of QBP-funded cases per hospital is based on historical volume levels at each hospital. The fixed fee is adjusted for each hospital based on its unique case-mix index (CMI) to account for the complexity in its overall patient population.

However, QBPs differ from most ABF reforms in that funding applies only to a very limited set of diagnoses and procedures, and they rely on the use of handbooks to encourage incorporation of best practices [12–15]. To create these handbooks for each QBP, the Ministry of Health and Long Term Care, in collaboration with partners such as Health Quality Ontario,

Cancer Care Ontario, and the Cardiac Care Network, established expert advisory panels with leading clinicians, scientists, and patients. They defined episodes of care for selected diagnoses or procedures, developed best practice recommendations for patient care, and suggested indicators to monitor for ongoing quality improvement. The resulting QBP Clinical Handbooks serve as a compendium of evidence and clinical consensus [11].

There is no mechanism in place to enforce adherence to the clinical pathways in the handbooks or to measure adherence; hospitals are paid via QBPs whether they follow the pathways or not, but the intent was that following the pathways would enable hospitals to deliver care for the amount paid by QBPs [7, 16].

To date, there has been no peer-reviewed evaluation of the overall effects of QBPs on key indicators of patient care. We took advantage of Ontario's data infrastructure to evaluate this new hospital payment model, focusing on system-level changes in measures of quality of care, access to care, and hospital coding behaviour for four QBPs including planned and unplanned surgical procedures, and medical diagnoses, selected *a priori* by our research team: (1) congestive heart failure, (2) hip fracture, (3) pneumonia, and (4) prostate cancer surgery.

## Methods

### Setting, context, and design

Hospital-based care in Ontario, Canada is publicly-funded. Ontario's 141 publicly-funded hospital corporations comprise 262 hospital sites [17], of which a majority receive QBP funding. Small hospitals (n = 55 with typically fewer than 2700 inpatients or day surgery cases per year in two of the last three years) and specialty hospitals—such as for mental health, children, chronic care, and rehabilitation—primarily receive funding through global budgets and have only implemented select QBPs (e.g. tonsillectomy) depending on their specific patient population (e.g. children). These hospitals are excluded from our analyses because they only perform a very small number, if any, of the diagnoses and/or procedures performed by QBP-funded hospitals [7].

Using population-based interrupted time series (ITS) analyses, and based on a dated pre-specified protocol and dataset creation plans held at ICES, we evaluated patients admitted to Ontario hospitals for four pre-specified QBPs. We selected these QBPs with input from health system decision makers and hospital leaders to represent a range of acute versus elective and surgical versus medical issues. This was further informed by our prior qualitative work which identified sources of potential variation in the extent to which, and the ways in which, hospitals responded to QBPs [7, 16]. We chose *a priori* to incorporate a 3 month transition period to allow time for any clinical changes in response to the funding model change to be implemented. We used an ITS design which is a robust quasi-experimental design that can be used to evaluate policy changes at the whole system- and population-level when randomization is infeasible. [18–22] The study interval depended on data availability and varied by QBP: congestive heart failure (April 2010-February 2017) launched April 2013; hip fracture (April 2012-February 2017) launched April 2014; pneumonia (April 2012-February 2017) launched April 2014; and prostate cancer surgery (April 2010-February 2017) launched April 2015. We chose these intervals to ensure at least 24 monthly data points pre-policy.

### Ethics approval

The use of data in this project was authorized under section 45 of Ontario's Personal Health Information Protection Act, which does not require review by a research ethics board.

## Study patients

We separately identified patients for each QBP cohort using inclusion and exclusion criteria detailed in each clinical handbook [12–15]. In short, cohorts for congestive heart failure and pneumonia were defined using specific qualifying hospital discharge diagnoses; hip fracture, using a combination of discharge diagnoses and procedures; and prostate cancer surgery, using specific procedure codes. We considered only admissions to hospitals that received funding for one of the QBPs under evaluation. Episodes of care had to be separated by at least 30 days (to exclude 30-day hospital readmissions). We excluded patients without a valid Ontario Health Insurance Plan (OHIP) number who could not be accurately followed in our data sets, and patients with missing demographic information (<0.1%).

## Data sources and quality measures

We used multiple linked health administrative databases to describe study patients and ascertain outcomes. Patient demographic information and vital status were obtained from the Registered Persons Database. Hospital diagnoses and procedures were obtained from the Canadian Institute for Health Information (CIHI) Discharge Abstract Database (CIHI-DAD). Emergency department admissions were captured using CIHI's National Ambulatory Care Reporting System (NACRS).

We described patients according to age at hospital admission, sex, neighbourhood income quintile, rurality of residence, Deyo-Charlson Comorbidity Index [23], and number of emergency department visits and hospitalization days in the year preceding qualifying admission.

Outcomes were assessed for each QBP in three domains:

1. Quality of care: i) death or return to hospital (i.e. unplanned presentation to emergency department or hospital admission within thirty days, among patients discharged alive and not transferred); ii) mean acute hospital length of stay (LOS); and iii) mean total LOS for entire episode of care including transfers;

2. Access to care: i) total volume of admissions; ii) proportion of patients aged 65 years or older; and iii) proportion of patients living in lowest neighborhood income quintile;

3. Coding behaviour: hospital discharge coding behavior as assessed by mean HBAM Inpatient Group (HIG) resource intensity weight. HIG weight is the Ontario-specific acute inpatient grouping methodology used to account for patients' clinical- and resource-utilization characteristics [24].

We selected these measures because policymakers hoped that QBPs would reduce length of stay in settings where it was longer than optimal without decreasing quality (i.e., outcomes such as deaths, return to hospital, or inappropriate coding). The expectation was also that shorter lengths of stay, as typically seen in other countries implementing ABF-like reforms, would facilitate greater throughput to increase total patient volume across the system, and that access to care would not be compromised by inequity across age and income [8, 25]. Socioeconomic status (SES) may contribute to inequalities in access to care, so we used neighborhood income quintile is an indicator of SES [26–28]. Prior research has shown that financial incentives associated with hospital funding reforms, such as QBPs, may alter coding behaviour to maximize reimbursement [29–32]. HIG weight is a measure of coding behaviour because it incorporates both case mix and the resource intensity of each patient care episode adjusted for patient characteristics. If upcoding is occurring we would expect to see changes in HIG weight. Thus, to the extent changes in HIG weight do not represent true abrupt changes in patient case mix, the HIG weight is one potential measure by which to evaluate effects on coding.

## Statistical analysis

For each outcome, we calculated monthly summaries, aggregated across hospitals (percent, mean, or raw count) and plotted them over time. For each QBP, we excluded three months of data following start of the funding change to account for a policy "transition" period [33]. We chose a three month transition *a priori*, postulating that it would take a fiscal quarter for any policy effects to occur. We accounted for seasonality by decomposing the data in trend, seasonal, and random components, and then removing the seasonal component [34].

We used segmented linear regression analysis of the seasonally-adjusted data. We used the forecast library in the R statistical software package to fit the model and used an automated stepwise selection procedure based on the Akaike Information Criterion (AIC) to include autoregressive terms accounting for the serial correlation [35, 36]. We used visual inspection of the observed and fitted data as well as residual plots to verify goodness of fit. Our model included fixed terms for pre-policy intercept and slope, intercept change at the time of policy (immediate difference in level following implementation of QBPs accounting for the postulated three-month transition period), and post-policy trend change (difference in the slope following implementation of QBPs). For the main results, we expressed the effect of QBP on each outcome as the counterfactual difference after 2 years, that is, the difference between the observed rate and the rate that would have occurred had QBPs been not implemented. This was estimated as the difference at 2 years post-implementation between the fitted post-implementation rates and the projected rates estimated from the pre-intervention intercept and slope. All analyses were performed at ICES (www.ices.on.ca) using linked, coded data. We used SAS v. 9.3 to prepare the monthly time series data for each outcome measure, and R version 3.4.4 to perform the regression analyses (nlme, car) and plot and compute the 95% confidence intervals for the counterfactual [37].

## Results

### Patient characteristics

S1–S4 Tables describe the overall characteristics for each cohort. The patient characteristics remained largely unchanged throughout the study period.

### Results from segmented regression analysis

The counterfactual estimates from the segmented regression analyses, representing the effect of QBPs on outcomes at 2 years post-implementation, are presented in Table 1. Figs 1–3 and S1–S4 Figs present the observed data and fitted values from the segmented regression analyses. The full results from the segmented regression analyses are presented in S5 and S6 Tables.

**Quality of care.** At 2 years post-implementation, the estimated percentage of hip fracture patients who returned to hospital or died within 30 days was higher by an absolute 3.13% (95% CI: 0.37% to 5.89%) than if QBPs had not been introduced. There was no change in LOS for hip fracture patients in comparison to the counterfactual (Table 1). For prostate cancer surgery patients, the increase in mean acute LOS over the counterfactual was 0.33 days (95% CI: 0.07 to 0.59) and mean total LOS 0.34 days (95% CI: 0.06 to 0.61). There were no other statistically significant changes observed for the LOS outcome (Fig 2 and S1 Fig).

**Access to care.** At 2 years post-implementation, the volume of patients admitted with congestive heart failure was higher by 80 patients (95% CI: 2 to 159) than if QBPs had not been introduced (Table 1). The volume of hip fracture admissions was lower by 138 patients (95% CI: -183 to -93). Percentage of admitted patients living in the lowest income quintile was

**Table 1. Estimated effect of implementation of QBPs on outcomes after 2 years, calculated as counterfactual difference from the segmented regression analysis (absolute difference, 95% confidence intervals).**

| | Hip Fracture | Congestive Heart Failure | Pneumonia | Prostate Cancer Surgery |
|---|---|---|---|---|
| %Percentage who were readmitted to hospital or died within 30 d | 3.13% (0.37 to 5.89) | 0.72% (-0.84 to 2.29) | 1.97% (-0.34 to 4.28) | 1.28% (-3.64 to 6.19) |
| Mean Acute Length of Stay, (days) | 0.31 (-0.69 to 1.3) | -0.16 (-0.61 to 0.3) | -0.18 (-0.61 to 0.25) | 0.33 (0.07 to 0.59) |
| Mean Total Length of Stay (days) | 0.71 (-1.31 to 2.72) | 0.16 (-0.46 to 0.78) | -0.23 (-0.62 to 0.16) | 0.34 (0.06 to 0.61) |
| Percentage% patients aged 65 years and older | 0.6% (-1.42 to 2.62) | -0.15% (-1.33 to 1.03) | 4.81% (-4.69 to 14.3) | 2.91% (-2.86 to 8.67) |
| Percentage% patients living in the lowest neighborhood income quintile | -1.65% (-4.31 to 1.01) | 1.89% (0.51 to 3.27) | 0.85% (-1.89 to 3.59) | -2.08% (-3.74 to -0.43) |
| Volume, n | -138 (-183 to -93) | 80 (2 to 159) | 258 (-90 to 607) | 19.7 (-42 to 81) |
| Mean HIG weight | 0.1% (-0.13 to 0.33) | 0.13% (-0.28 to 0.54) | -0.07% (-0.24 to 0.11) | 0.01% (-0.05 to 0.06) |

Abbreviations: HIG, Health Based Allocation Model (HBAM) Inpatient Group (HIG)

higher (1.89%; 95% CI: 0.51% to 3.27%) for those diagnosed with congestive heart failure and decreased for those with prostate cancer surgery (-2.08%; 95% CI: -3.74% to -0.43%).

**Hospital coding behaviour.**   We observed no statistically significant changes in mean HIG weight for any of the four cohorts (S4 Fig).

## Interpretation

### Summary of findings

For the seven outcomes across the four diagnoses and/or procedures we studied, we compared the observed against expected counterfactual findings across outcomes in the domains of quality, access, and coding behaviour and found an inconsistent and generally weak response to the QBP funding reform at the system-level. In general, QBPs did not appear to result in changes to prevailing trends in return to hospital or death. Contrary to expectations, LOS increased slightly for prostate cancer surgery. Counterintuitively, despite no change in LOS, QBP funding was associated with a decrease in overall volume of admissions for hip fracture surgeries and a small absolute increase (of 3%) in the monthly percentage of hip fracture patients who returned to hospital or died within 30 days of discharge. We observed small increases in the percentage of patients admitted with congestive heart failure and decreases for prostate cancer surgery residing in the lowest neighborhood income quintile (<2%).

### Explanation of findings

To our knowledge, there is no published quantitative research on the broad effects of the implementation of the QBP funding reform policy. Prior qualitative analyses of QBPs has revealed challenges associated with implementation of this complex hospital funding reform policy [7, 16, 38].

Previous studies found mixed reactions after other types of hospital funding reform [25, 39]. For example, evaluation of a limited experiment with activity-based funding in British Columbia, Canada, showed small decreases in volume, small increases in patients' length of stay, and no changes in measures of quality (i.e., unplanned readmissions and in-hospital mortality) [25]. Conversely, a systematic review of ABF affirmed that transition to activity-based funding initially decreased length of stay in the US and internationally, and also found important policy- and clinically-relevant changes, including substantial increases in admissions to post-acute care following hospitalization [8].

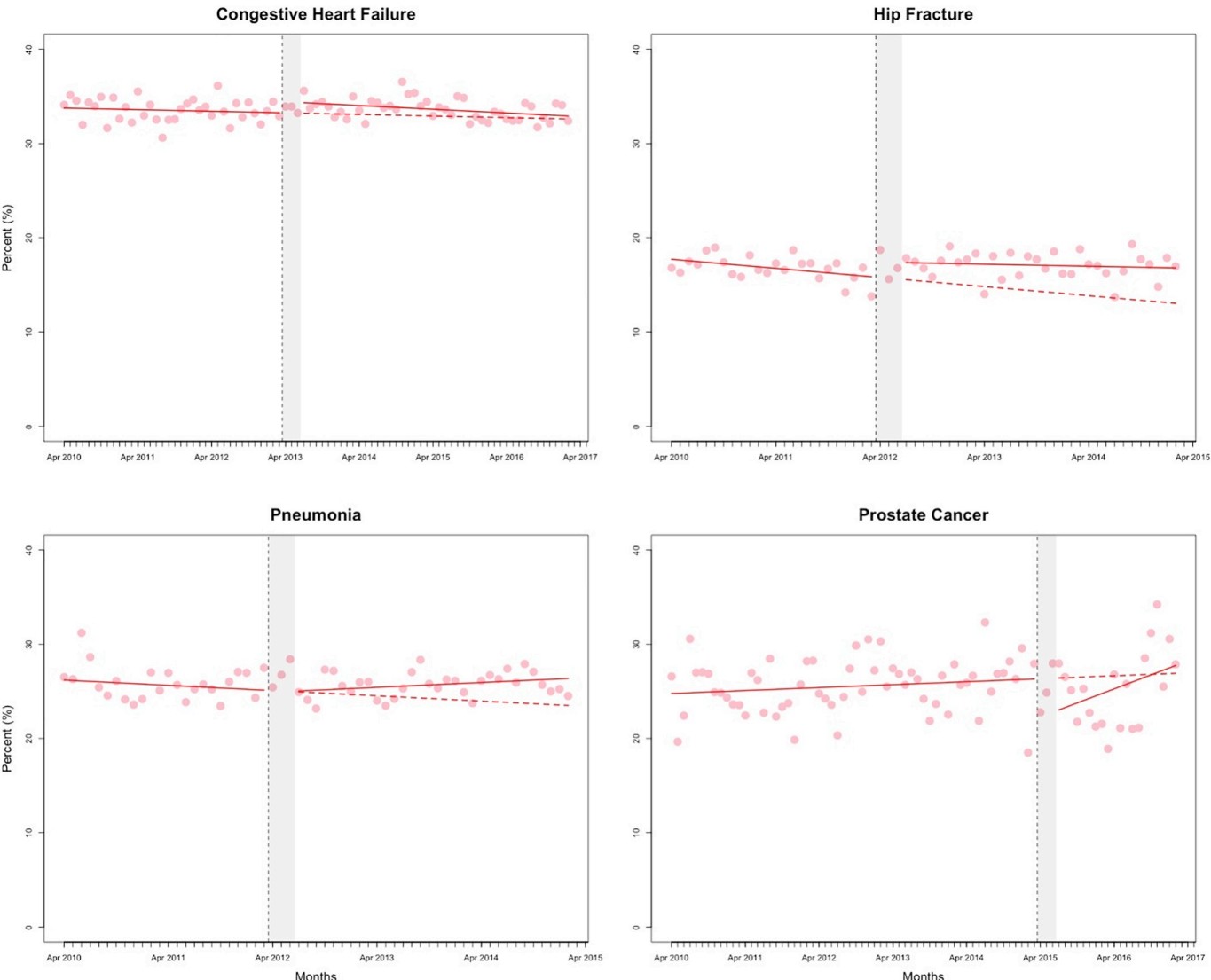

**Fig 1. Percent of patients returned to hospital or died.** Red solid line represents the fitted model. The red dashed line represents the counterfactual (i.e. if no policy change occurred). The vertical dashed line represents the date of policy change. The grey shaded area represents the three months of "transition" period. Data are seasonally adjusted.

Unintended consequences typically associated with ABF-like reforms include, for example, patients being discharged "sicker and quicker" to post-acute care facilities or home, and "upcoding", which may be appropriate if it represents more accurate coding, or inappropriate [8]. We did not see the decrease in LOS that might have been precipitated by accelerated discharge. Nor did we observe changes in coding behaviour for the QBPs studied.

The slight variation we observed in effects—some positive, others negative—may be partly explained by our prior qualitative work, in which we observed variation in response to the reform related to complexity of changes required, internal capacity for organizational change, and availability and appropriateness of supports to manage change [16]. It may also simply represent noise rather than signal. Our goal in this paper was not to understand quantitatively

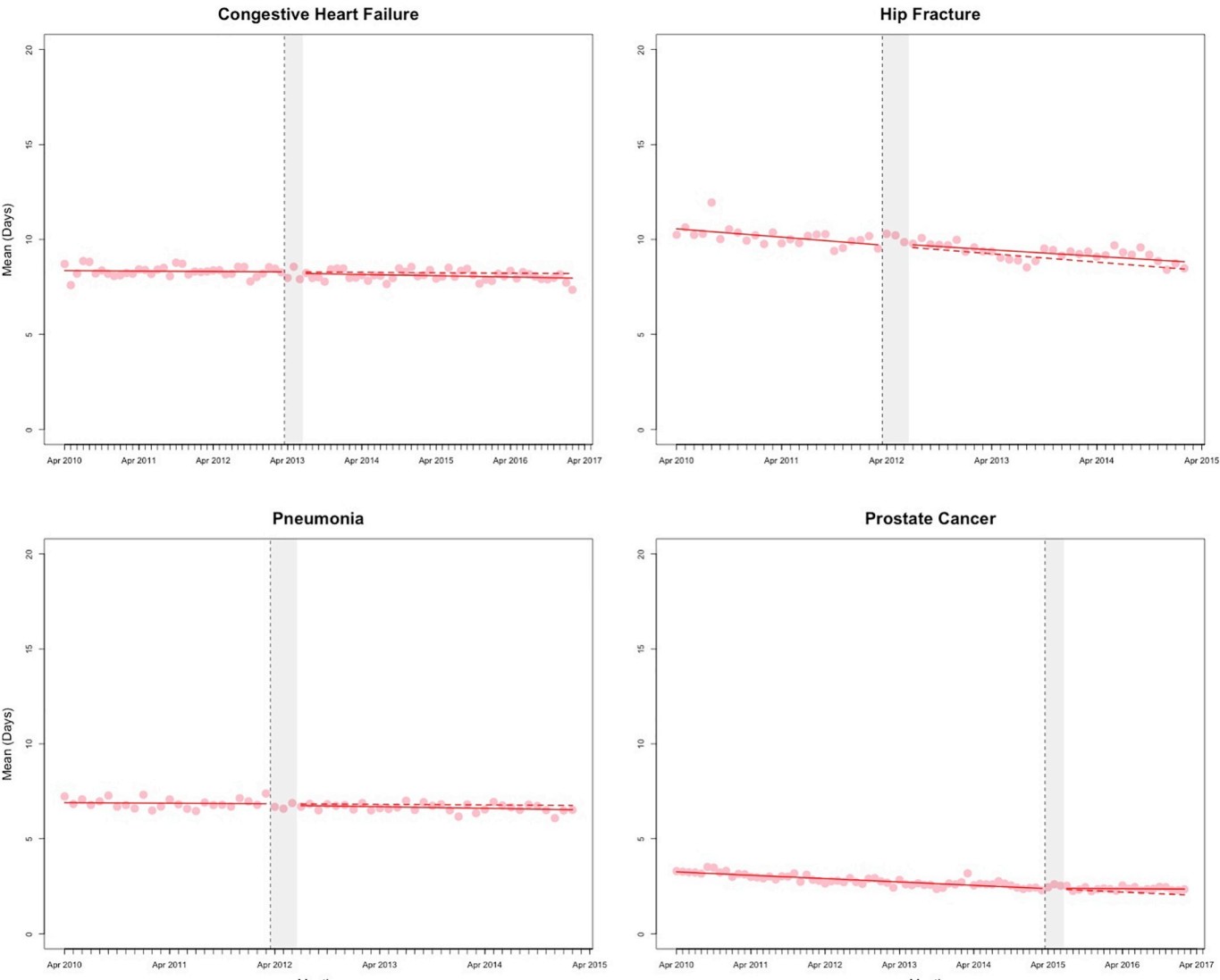

**Fig 2. Mean acute length of stay for the episode of care (days).** Red solid line represents the fitted model. The red dashed line represents the counterfactual (i.e. if no policy change occurred). The vertical dashed line represents the date of policy change. The grey shaded area represents the three months of "transition" period. Data are seasonally adjusted.

the variation in responsiveness by hospital, but to evaluate the system-level effects of the jurisdictional policy change, as this is the level at which 'success' of the policy reform must ultimately be judged.

The lack of large-scale meaningful changes in association with Ontario's shift to QBP funding is perhaps not surprising. Funding reforms may not be necessary or effective when desirable changes are already occurring. For example, hospitals were already under long-standing pressures to reduce length of stay and may have reached a floor, which may explain why further financial pressure from QBPs had little effect. Similarly, hospitals may have also lacked effective incentives or supports to address readmissions, since, unlike activity-based funding reforms in other countries, Ontario's QBP funding reform did not financially disincentivize return to hospital nor link funding to care outcomes.

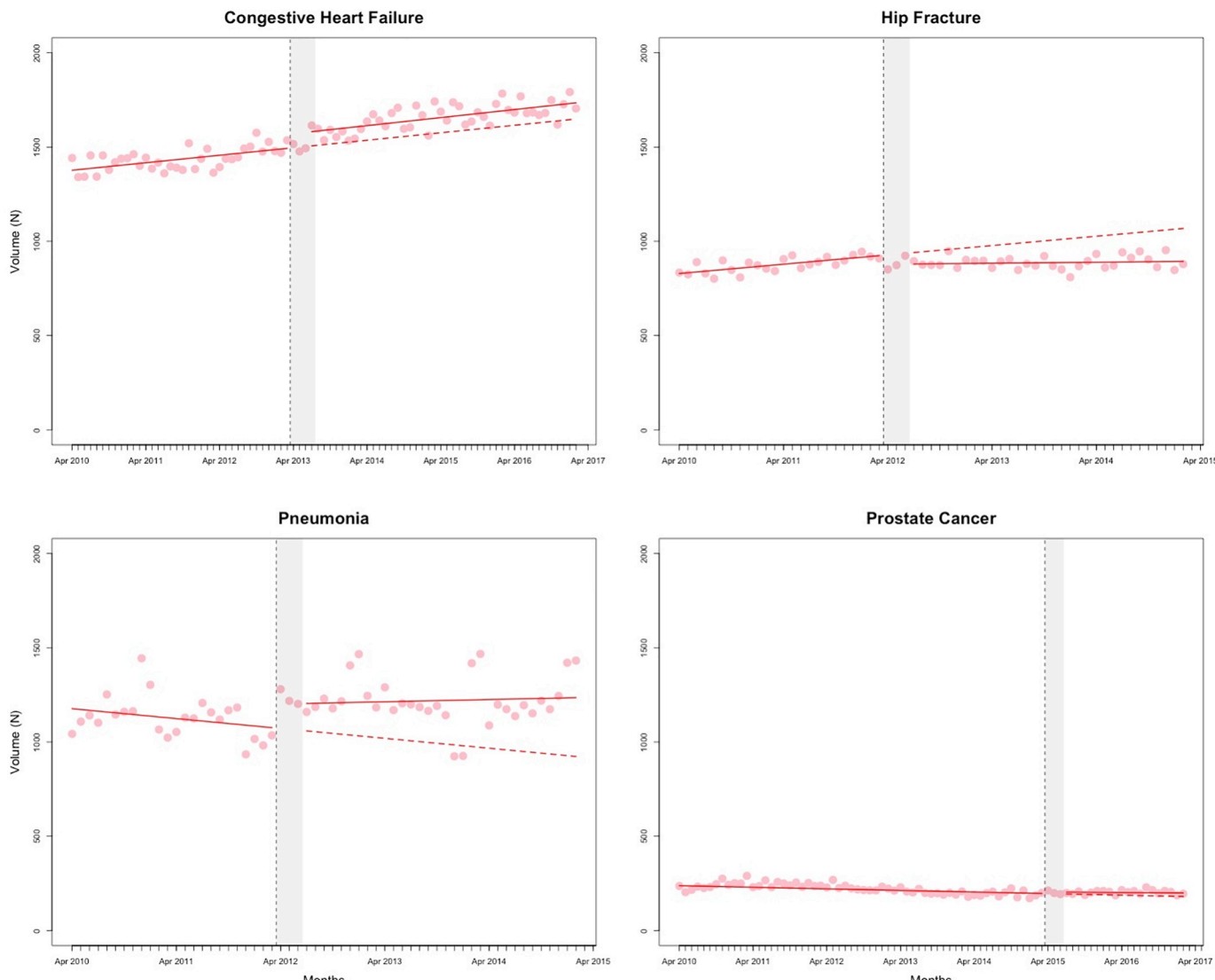

**Fig 3. Total volume of admissions.** Red solid line represents the fitted model. The red dashed line represents the counterfactual (i.e. if no policy change occurred). The vertical dashed line represents the date of policy change. The grey shaded area represents the three months of "transition" period. Data are seasonally adjusted.

## Limitations

Our study has several important limitations that are common to observational studies of policy changes. Teasing apart the effect of QBPs in the presence of multiple system-level changes is challenging. First, other initiatives to improve patient care and/or control costs may have overlapped with the timing of QBP implementation. Specific initiatives that we are aware of included passage in 2010 of the Excellent Care for All Act (ECFA) [40], the introduction of Health Based Allocation Model hospital funding reforms in April 2012 [24], and the introduction of Community Health Links in December 2012 [41] (S5–S7 Figs, S7 Table). Visual inspection of data points around the timing of introduction of these initiatives however suggests that they are unlikely to have had a major impact on the outcomes studied in our analyses. A possible exception is the introduction of Community Health Links in December 2012: due to its timing close to that of the QBPs for congestive heart failure (CHF) in April 2013, it

is difficult to independently assess the effect of the QBPs for this condition. Undetected confounding is always possible in any uncontrolled study. Policies aimed at improving health care are constantly being tinkered with, which may influence any particular intervention, such as QBPs, in ways not easily detected. Second, given the nearly ubiquitous implementation of QBP in Ontario, we did not identify suitable contemporaneous comparators in this study, which could have strengthened the inferences drawn. Although one of the ways to optimize an ITS is to add negative controls, we did not add these because the goal of the reform was to effect broad change across the entire system, resulting in only a small number of unique and, therefore, non-comparable hospitals being exempted from implementing QBPs (i.e., very small hospitals with few beds and/or those with unique targeted populations). Third, because our analyses were restricted to QBP-funded hospitals rather than all hospitals, we cannot be certain that our results are generalizable to the whole system; however, the proportion of QBP procedures occurring outside of QBP-funded hospitals is low (<11%). Fourth, examining a broader range of outcomes (e.g. extent to which patient care is aligned with evidence-based care processes described in QBP clinical pathways; reduction in inter-hospital variation in care, cost, and wait-times) may be more sensitive to, or reveal different effects of QBPs on, patient care and outcomes, care providers, and the health care system as a whole. Fifth, it is noteworthy that QBPs are unique to Ontario, making generalizability to other jurisdictions (both within and outside Canada) difficult to assess. Although somewhat similar in design to ABF reforms elsewhere, critical differences include the absence of financial disincentives for readmission with QBPs; a smaller and less ubiquitous funding scope limited to fewer priority diagnoses and procedures than ABF reforms elsewhere. However, this study is relevant to health system funding reforms that attempt to improve quality while also cutting costs, though contextual factors that influence the linkage between quality and cost are difficult to capture in relatively simple funding reforms [42]. Sixth, we did not assess how QBPs impacted hospitals' finances, so we cannot make any inferences about whether increases or decreases in hospitals' budgets affected patient care and/or outcomes for the QBPs we evaluated. Seventh, there may be benefits or harms of QBPs that we did not measure, or other policy objectives that may have been met, such as those related to total cost per episode-of-care or cost to the system overall. We were careful to limit our conclusions to only the QBPs and outcomes we evaluated to avoid being overbroad.

## Conclusion

We found mixed and generally very small effects on quality of care, access to care, and coding behaviour, across the four QBPs we studied. We speculate that challenges with implementing the best practice pathways featured in the QBP handbooks, together with progressive controls on hospital expenditures, and a worsening overall fiscal picture in Ontario coincident with QBP implementation, may have led to inconsistent and weak signals. Further experimentation with funding reform as a potential mechanism to improve outcomes might yield greater impact if focused on specific diagnoses and procedures in which suboptimal process or outcome measures are well-established and for which efforts to improve outcomes by other means have been inadequate.

## Supporting information

**S1 Fig. Mean length of stay for entire episode (days).** Red solid line represents the fitted model. The red dashed line represents the counterfactual (i.e. if no policy change occurred). The vertical dashed line represents the date of policy change. The grey shaded area represents

the three months of "transition" period. Data are seasonally adjusted.
(TIFF)

**S2 Fig. Percent change in patients over 65 (%).** Red solid line represents the fitted model. The red dashed line represents the counterfactual (i.e. if no policy change occurred). The vertical dashed line represents the date of policy change. The grey shaded area represents the three months of "transition" period. Data are seasonally adjusted.
(TIFF)

**S3 Fig. Percent change in patients living in the lowest neighborhood income quintile.** Red solid line represents the fitted model. The red dashed line represents the counterfactual (i.e. if no policy change occurred). The vertical dashed line represents the date of policy change. The grey shaded area represents the three months of "transition" period. Data are seasonally adjusted.
(TIFF)

**S4 Fig. Mean HIG weight.** Red solid line represents the fitted model. The red dashed line represents the counterfactual (i.e. if no policy change occurred). The vertical dashed line represents the date of policy change. The grey shaded area represents the three months of "transition" period. Data are seasonally adjusted.
(TIFF)

**S5 Fig. Percent of patients returned to hospital or died with competing initiatives.** Red solid line represents the fitted model. The red dashed line represents the counterfactual (i.e. if no policy change occurred). The vertical dashed line represents the date of policy change. The grey shaded area represents the three months of "transition" period. Competing initiatives are outlined in the legend. Data are seasonally adjusted.
(TIFF)

**S6 Fig. Mean acute length of stay for the episode of care (days) with competing initiatives.** Red solid line represents the fitted model. The red dashed line represents the counterfactual (i.e. if no policy change occurred). The vertical dashed line represents the date of policy change. Competing Interventions are outlined in the legend. The grey shaded area represents the three months of "transition" period. Competing Initiatives are outlined in the legend. Data are seasonally adjusted.
(TIFF)

**S7 Fig. Total volume of diagnoses/procedures with competing initiatives.** Red solid line represents the fitted model. The red dashed line represents the counterfactual (i.e. if no policy change occurred). The vertical dashed line represents the date of policy change. The grey shaded area represents the three months of "transition" period. Competing Initiatives are outlined in the legend. Data are seasonally adjusted.
(TIFF)

**S1 Table. Cohort characteristics for congestive heart failure patients included in the analysis.**
(DOCX)

**S2 Table. Cohort characteristics for hip fracture patients included in the analysis.**
(DOCX)

**S3 Table. Cohort characteristics for pneumonia patients included in the analysis.**
(DOCX)

**S4 Table. Cohort characteristics for prostate cancer patients included in the analysis.**
(DOCX)

**S5 Table. Estimated coefficients from the interrupted time series analysis on the impact of quality-based procedure policy (% patients returned to hospital or died, mean acute length of stay, total volume).**
(DOCX)

**S6 Table. Estimated coefficients from the interrupted time series analysis on the impact of quality-based procedure policy (% change in patients over 65, % change in patients living in lowest neighborhood income quintile, mean HIG weight.**
(DOCX)

**S7 Table. Overlapping initiatives.**
(DOCX)

**S1 File.**
(ZIP)

## Acknowledgments

We thank Michael Law, University of British Columbia, for insightful comments on an earlier draft of the manuscript.

## Author Contributions

**Conceptualization:** Alvin Ho-ting Li, Karen S. Palmer, J. Michael Paterson, Adalsteinn Brown, Noah Ivers.

**Data curation:** Alvin Ho-ting Li, J. Michael Paterson, Anjie Huang.

**Formal analysis:** Alvin Ho-ting Li, Karen S. Palmer, Monica Taljaard, J. Michael Paterson, Anjie Huang, Lauren Lapointe-Shaw, Noah Ivers.

**Funding acquisition:** Adalsteinn Brown, Noah Ivers.

**Investigation:** Alvin Ho-ting Li, Karen S. Palmer, J. Michael Paterson, Anjie Huang, Noah Ivers.

**Methodology:** Alvin Ho-ting Li, Monica Taljaard, J. Michael Paterson, Adalsteinn Brown, Noah Ivers.

**Project administration:** Alvin Ho-ting Li, Karen S. Palmer, Noah Ivers.

**Software:** Alvin Ho-ting Li, J. Michael Paterson, Anjie Huang.

**Supervision:** Alvin Ho-ting Li, Karen S. Palmer, Noah Ivers.

**Validation:** Alvin Ho-ting Li, J. Michael Paterson, Noah Ivers.

**Visualization:** Alvin Ho-ting Li.

**Writing – original draft:** Alvin Ho-ting Li, Karen S. Palmer.

**Writing – review & editing:** Alvin Ho-ting Li, Karen S. Palmer, Monica Taljaard, J. Michael Paterson, Adalsteinn Brown, Anjie Huang, Husayn Marani, Lauren Lapointe-Shaw, Daniel Pincus, Marian S. Wettstein, Girish S. Kulkarni, David Wasserstein, Noah Ivers.

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
