## [Decision Letter · Decision Letter 0]

10 Dec 2019

PONE-D-19-27876

Effects of Quality-based Procedure hospital funding reform in Ontario, Canada: An Interrupted Time Series Study

PLOS ONE

Dear Dr Li,

Thank you for submitting your manuscript to PLOS ONE. After careful consideration, we feel that it has merit but does not fully meet PLOS ONE’s publication criteria as it currently stands. Therefore, we invite you to submit a revised version of the manuscript that addresses the points raised during the review process.

Both reviewers indicated the potential of the manuscript but also highlighted some points that need to be addressed. In particular, both raised the point that QBP may not be generally known sufficiently to many readers. Providing additional information on QBP is therefore important and needs to be carefully addressed in your revisions. I would also like to ask you to address the concerns about not following the PLOS ONE policy made by one of the reviewers.

We would appreciate receiving your revised manuscript by Jan 24 2020 11:59PM. To enhance the reproducibility of your results, we recommend that if applicable you deposit your laboratory protocols in protocols.io, where a protocol can be assigned its own identifier (DOI) such that it can be cited independently in the future. For instructions see: http://journals.plos.org/plosone/s/submission-guidelines#loc-laboratory-protocols

We look forward to receiving your revised manuscript.

Kind regards,

Hans-Peter Brunner-La Rocca, M.D.

Academic Editor

PLOS ONE

Journal Requirements:

2. In the ethics statement in the manuscript and in the online submission form, please provide additional information about the patient records used in your retrospective study.

Specifically, please ensure that you have discussed whether all data were fully anonymized before you accessed them and/or whether the IRB or ethics committee waived the requirement for informed consent.

If patients provided informed written consent to have data from their medical records used in research, please include this information.

'We have read the journal's policy and the authors of this manuscript have the following competing interests: NI, AB, KP, HM report funding from Ontario Strategy for Patient Oriented Research Support Unit (OSSU) Impact Award during the conduct of the study; OSSU was in turn funded by the Canadian Institutes of Health Research (CIHR) and the Government of Ontario. NI reports support from a CIHR New Investigator Award and from the Department of Family and Community Medicine, University of Toronto, unrelated to this work. AB is a former senior official in the Ontario Government and serves on the current Premier’s Council on Health. AL reports support from a CIHR Fellowship Award.'

5. Your ethics statement must appear in the Methods section of your manuscript. If your ethics statement is written in any section besides the Methods, please move it to the Methods section and delete it from any other section. Please also ensure that your ethics statement is included in your manuscript, as the ethics section of your online submission will not be published alongside your manuscript.

Reviewers' comments:

Reviewer's Responses to Questions

**Comments to the Author**

1. Is the manuscript technically sound, and do the data support the conclusions?

Reviewer #1: Partly

Reviewer #2: Yes

2. Has the statistical analysis been performed appropriately and rigorously? 

Reviewer #1: I Don't Know

Reviewer #2: Yes

3. Have the authors made all data underlying the findings in their manuscript fully available?

Reviewer #1: No

Reviewer #2: Yes

4. Is the manuscript presented in an intelligible fashion and written in standard English?

Reviewer #1: Yes

Reviewer #2: Yes

5. Review Comments to the Author

Reviewer #1: The study attempts to estimate the effects of an intervention designed to improve the quality of patient care provided by hospitals in the province of Ontario Canada. The intervention consists of the introduction of Quality-based Procedures (QBPs) for the provision of care for inpatients being treated for congestive heart failure, hip fracture, pneumonia, and prostate cancer. The authors focus on monthly data on various measures of the quality of care of those patients who were admitted to hospital to treat these conditions, both before the introduction of the intervention and afterwards. The authors also examined the total volume of cases and the fraction of cases that were from more compromised patients (those from low income neighbourhoods and those 65+ years of age). This used an interrupted time series analysis with various adjustments for seasonality and AR errors. There was little if any change in the pre policy trend line after the policy was introduced, leading the authors to conclude that these policies were largely ineffective.

This was a good study with potential. I have two substantive comments on the methods and one on the data disclosure (which I am required to address as a referee).

First, the authors disclose that i) the QBP consisted of a fixed dollar remuneration for the entire episode of care, and the provision of information on “best practices” for the treatment of the respective conditions and ii) there were no financial penalties for bad outcomes or non-adherence.

Other features of the QBP are not explained. For instance, the authors speak of there being a “fixed volume” component to the QBP: “fixed fee and fixed volume, for each QBP procedure or diagnosis.” It is unclear what this means. Does it mean for instance that the government would pay only for a certain volume of cases? If so, what would happen to the funds should the number of patients treated fall below the maximum number allowed? Would this money flow back to the hospital?

If a patient whose treatment falls under the QBP rubric is discharged but then re-admitted to the hospital due to complications, how is the cost of the readmission covered? Is this supposed to come from the fixed remuneration for that patient?

What measures were present, if any, to prevent hospitals from cream skimming? Is this even possible? Could hospitals have any influence on the disease severity and volume of patients seeking care? Is the remuneration per patient adjusted for patient characteristics or the complexity of the condition?

It would also be helpful to understand the ability of hospital management to direct the clinical care of patients in their hospitals. How much discretion did they have?

Information on the nature of the intervention and how it would have affected clinical decision making in the hospital might go some ways to explaining the results.

Second, before and after designs, such as the kind used in this study, face the challenge of distinguishing the effect of the intervention from the effects of other interventions that were introduced over the sample period. The authors describe some of these other interventions but are of the opinion that they can be safely ignored because i) none were introduced at the exact same time as the QBP policy and ii) those that were introduced before the policy would have effects that “would have been captured in the secular trend” and iii) those that were introduced after the policy would have effects that “did not affect the entire system in a reliable fashion for the procedures, diagnoses, or outcomes under investigation, or did not have a specific time when changes in the outcomes measured might have been expected.”

I don’t find these arguments persuasive. There are multiple ways that the QBP effect could be obscured by other effects. There is no requirement for instance that any policies introduced prior to the intro of the QBP would have effects that would combine and result in a continuous pre policy linear time trend. The statement that policies introduced after QBP had no impact on trends because one cannot predict when changes in outcomes would have occurred suggests that indeed these other policies could have had some impact.

I would address this issue in two ways. First, provide some more info on the other policies that were introduced over the sample period and provide some justification that they indeed have no material impact on the trends in the outcomes under investigation. This could be contained in an appendix.

Second it may be helpful as well to run a structural break test on the time series data, allowing the data to determine if there were any structural breaks at any point in the time series data. The references contained in the following study may be helpful

https://link.springer.com/article/10.1007/s10614-011-9271-1

In the cases where the authors have detected a break in trend (such as for hip fracture mortality), it would be useful to see if the automated structural break detection algorithm also detects this. This would lend some confidence that the break really is due to the QBP policy.

The referee form also requires that I comment on the steps taken by the researchers to make their data available to other researchers so as to allow for independent replication. The authors have elected to keep their “source data” – both the individual level data and the aggregated data – confidential. They do display graphs of the trends of the deseasonalized data. But do not present the actual values of the aggregated data in either raw or deseasonalized forms. The claimed justification for the secrecy is a provision of Ontario’s “Personal Health Information Privacy Act” which evidently prohibits disclosure. My sense is that the study authors do not justify why disclosure is not permitted. I am very interested to understand exactly how the disclosure of data with personal identifiers removed could violate the Act. Exactly what is the privacy breach for releasing the individual level data with personal identifiers removed? How would the Act be violated if the aggregated unadjusted data were to be released?

Reviewer #2: General Comments:

1. QBP's and quality measures/clinical handbooks. The content of the QBP quality handbooks will not be familiar to many readers. Can the authors provide basic information on the content, scope, quality/rigor, and specificity of the clinical guidance in the QBP quality handbooks, perhaps in an online supplement? To what extent would the quality of care recommendations differ from the standard of care already in place in Canadian teaching hospitals or large community hospitals? How evidence-based and specific are the recommendations? Was there ever a realistic prospect that providing information in QBP handbooks could measurably improve quality of care, with or without an enforcement mechanism in QBP administration?

2. Is there a reason why QBP implementation would be expected to affect the number of admissions for congestive heart failure, pneumonia, hip fracture, and even prostatectomy? I would think that the number of admissions would be driven primarily by the disease incidence, rather that hospital capacity or funding models. More elective procedures, such as non-cancer surgery e.g. knee and hip joint replacement, might be more sensitive to changes in funding models.

3. While implementation of QBPs did not affect the outcomes the authors evaluated, it is possible that the funding model did achieve other policy objectives. To the extent that QBPs are intended to limit the financial risk of the provincial funder associated with shortfalls in annual hospital budgets, it is possible that QBP funding did achieve some policy objectives. Further, it is possible that episode-of-care funding incentivized hospitals to provide more efficient and less costly care for these episodes. The investigators would not have detected these efficiencies, since they would not likely be reflected in changes in clinical outcome, or even general resource utilization measures such as length of stay.

Specific Comments:

1. Abstract "Patients from the lowest income neighborhoods increased slightly..." will be a confusing statement to people not familiar with SES income quintiles. Perhaps state something like "The proportion of patients admitted for congestive heart failure who lived in the lowest neighborhood income quintile increased.." or something like that which more clearly indicates the direction of any change in access/SES of target population.

2. Introduction 3rd paragraph "Funding was carved out of hospitals' global budgets and then reallocated…as a fixed fee" is not entirely correct, since funding per episode of care is a product of a fixed fee and the CMI of the hospital (a hospital-level attribute based on historical patient case mix for similar admissions). The amount of funding for the same QBP varies significantly from hospital to hospital based on CMI.

3. Page 7 What is "responsiveness" to QBP's? ("prior qualitative work which identified sources of potential variation in responsiveness to QBP's")

4. Page 9 explanation of access to care measures. More explanation about why the "proportion of patients living in lowest neighborhood income quintile" is a measure of access would help average readers understand this concept.

5. Page 9 coding behaviour. Can you explain why mean HBAM HIG RIW trend is a measure of coding "behaviour"? How can you know if changes in RIW's do not represent true abrupt changes in patient case mix?

6. Page 11 results 2nd paragraph "...the effect of QBPs on outcomes 2 years after implementation..." (don't need to say "after 2 years post-implementation"). Same comment applies throughout eg 1st sentence 3rd paragraph of results.

7. Page 14 discussion. "The slight variation we observed in effects--some results went up, others down...". These changes may still reflect noise rather than signal. I would not necessarily assume any causal association with the exposure. Other explanations do include random chance, bias, other secular trends, etc.

8. Page 15 "...the goal of the reform was to effect broad change..." (not affect)

6. PLOS authors have the option to publish the peer review history of their article (what does this mean?). If published, this will include your full peer review and any attached files.

Reviewer #1: No

Reviewer #2: Yes: David Urbach

---

## [Author Response · Author response to Decision Letter 0]

30 Jan 2020

Authors’ Response to 1:

We have carefully checked that the manuscript now meets PLOS ONE’s style requirements for title page, file naming, and body of manuscript. 

2. In the ethics statement in the manuscript and in the online submission form, please provide additional information about the patient records used in your retrospective study. Specifically, please ensure that you have discussed whether all data were fully anonymized before you accessed them and/or whether the IRB or ethics committee waived the requirement for informed consent. If patients provided informed written consent to have data from their medical records used in research, please include this information.

Authors’ Response to 2: 

We have updated our ethics statement to read as follows:

“The use of data in this project was authorized under section 45 of Ontario’s Personal Health Information Protection Act, which does not require review by a research ethics board.”

Authors’ Response to 3:

We’ve consulted with our organization and we now write:

“The dataset from this study is held securely in coded form at ICES. While data sharing agreements prohibit ICES from making the dataset publicly available, access may be granted to those who meet pre-specified criteria for confidential access, available at www.ices.on.ca/DAS. The full dataset creation plan and underlying analytic code are available from the authors upon request, understanding that the computer programs may rely upon coding templates or macros that are unique to ICES and are therefore either inaccessible or may require modification.”

'We have read the journal's policy and the authors of this manuscript have the following competing interests: NI, AB, KP, HM report funding from Ontario Strategy for Patient Oriented Research Support Unit (OSSU) Impact Award during the conduct of the study; OSSU was in turn funded by the Canadian Institutes of Health Research (CIHR) and the Government of Ontario. NI reports support from a CIHR New Investigator Award and from the Department of Family and Community Medicine, University of Toronto, unrelated to this work. AB is a former senior official in the Ontario Government and serves on the current Premier’s Council on Health. AL reports support from a CIHR Fellowship Award.'

Authors’ Response 4a: These competing interests do not alter our adherence to PLOS ONE policies. Our data sharing agreements allow access to any researcher who meets pre-specified criteria for confidential access, available at www.ices.on.ca/DAS. 

Authors’ Response 4b: Our revised competing interests statement now reads as follows, and we now also include the updated competing interests in the body of our cover letter, as requested.

“We have read the journal's policy and the authors of this manuscript have the following competing interests: NI, AB, KP, HM report funding from Ontario Strategy for Patient Oriented Research Support Unit (OSSU) Impact Award during the conduct of the study; OSSU was in turn funded by the Canadian Institutes of Health Research (CIHR) and the Government of Ontario. NI reports support from a CIHR New Investigator Award and from the Department of Family and Community Medicine, University of Toronto, unrelated to this work. AB is a former senior official in the Ontario Government and serves on the current Premier’s Council on Health. AL reports support from a CIHR Fellowship Award. These competing interests do not alter our adherence to PLOS ONE policies. Our data sharing agreements allow access to researchers who meets pre-specified criteria for confidential access, available at www.ices.on.ca/DAS.” 

5. Your ethics statement must appear in the Methods section of your manuscript. If your ethics statement is written in any section besides the Methods, please move it to the Methods section and delete it from any other section. Please also ensure that your ethics statement is included in your manuscript, as the ethics section of your online submission will not be published alongside your manuscript.

Authors’ Response to 5:

Our ethics statement now appears in the Methods section of the manuscript, and does not appear in any other section. 

Reviewers' comments and Authors’ responses:

Reviewer #1: 

6. The study attempts to estimate the effects of an intervention designed to improve the quality of patient care provided by hospitals in the province of Ontario Canada. The intervention consists of the introduction of Quality-based Procedures (QBPs) for the provision of care for inpatients being treated for congestive heart failure, hip fracture, pneumonia, and prostate cancer. The authors focus on monthly data on various measures of the quality of care of those patients who were admitted to hospital to treat these conditions, both before the introduction of the intervention and afterwards. The authors also examined the total volume of cases and the fraction of cases that were from more compromised patients (those from low income neighbourhoods and those 65+ years of age). This used an interrupted time series analysis with various adjustments for seasonality and AR errors. There was little if any change in the pre policy trend line after the policy was introduced, leading the authors to conclude that these policies were largely ineffective.

This was a good study with potential. I have two substantive comments on the methods and one on the data disclosure (which I am required to address as a referee).

First, the authors disclose that i) the QBP consisted of a fixed dollar remuneration for the entire episode of care, and the provision of information on “best practices” for the treatment of the respective conditions and ii) there were no financial penalties for bad outcomes or non-adherence.

Other features of the QBP are not explained. For instance, the authors speak of there being a “fixed volume” component to the QBP: “fixed fee and fixed volume, for each QBP procedure or diagnosis.” It is unclear what this means. Does it mean for instance that the government would pay only for a certain volume of cases? If so, what would happen to the funds should the number of patients treated fall below the maximum number allowed? Would this money flow back to the hospital?

Authors’ Response to 6:

Yes, government pays for only a certain volume of QBP-funded cases per hospital, based on historical volume levels at each hospital. However, we know from prior qualitative research (URLs below) that hospitals could “borrow” from their own global budget to cover the costs of any diagnoses/procedures in excess of those funded by QBPs. We don't know the extent to which this borrowing happens, other than it does. Conversely, if the number of patients treated falls below the maximum allowed, the funds are ostensibly returned to government, but we have no evidence of the extent to which this happens either. 

Our statement in the manuscript remains accurate as written, “Funding was carved out of hospitals’ global budgets and then reallocated to hospitals at the start of the relevant fiscal year as a fixed fee and fixed volume, for each QBP procedure or diagnosis.” Although Reviewer #1 asks good questions, we feel that revising the manuscript to drill down to this level of detail about the flow of funds back and forth between government and hospitals goes beyond what is required in this manuscript on quantitative effects. We discuss this in our prior qualitative research publications, including these two, both of which we cite early in the manuscript at references 7 and 12 for those interested in learning more about the implementation of QBPs: 

https://journals.plos.org/plosone/article?id=10.1371/journal.pone.0191996

https://health-policy-systems.biomedcentral.com/articles/10.1186/s12961-018-0353-6

. 

7. If a patient whose treatment falls under the QBP rubric is discharged but then re-admitted to the hospital due to complications, how is the cost of the readmission covered? Is this supposed to come from the fixed remuneration for that patient?

Authors’ Response to 7:

If readmission occurs within 30 days of discharge due to complications, treatment costs are not funded from the fixed remuneration for the original admission, but rather from the global budget. In this sense, QBPs are not a bundled payment that would include the costs of readmission. If a patient is readmitted after 30 days, even for the same diagnosis/procedure, or for complications arising from the original admission, it is considered a new admission, funded anew through additional QBP funds, separate from the original amount. 

8. What measures were present, if any, to prevent hospitals from cream skimming? Is this even possible? Could hospitals have any influence on the disease severity and volume of patients seeking care? Is the remuneration per patient adjusted for patient characteristics or the complexity of the condition?

Authors’ Response to 8:

Q 8a: “What measures were present, if any, to prevent hospitals from cream skimming? Is this even possible?”

A 8a: We know of no specific measures to prevent adverse risk selection, aka cream skimming (i.e. selection of high-value, lower-severity, low-cost patients to enhance profitability or reputation). To examine the extent to which this might be occurring, we evaluated 3 variables related to adverse risk selection: 1) patients’ income quintile; 2) proportion of admissions over age 65; 3) Charlson Score (co-morbidity index, an indicator of severity of illness). We found no evidence of adverse selection for these indicators, meaning no decline after QBPs in the proportion of poorer, older, or sicker patients being admitted. It does not appear, based on the outcomes we examined, that the hospital system, as a whole, was influencing patient selection, though it is possible that individual hospitals may have done so. We did not report data at the level of individual hospitals because our interest was in effects at the system-level. 

Q 8b: “Could hospitals have any influence on the disease severity and volume of patients seeking care?” 

A 8b: Hospitals could not influence the disease severity or volume of patients “seeking” care (as the reviewer asks), because any patient is free to seek care in any Canadian hospital. Hospitals could, theoretically, influence who is treated or admitted to hospital for care. We did not study, for example, transport diversions, but it seems unlikely that a diversion for unplanned care would occur based only on a suspected diagnosis as communicated by a paramedic en route in an ambulance. Nor did we study wait list manipulation for planned care, by which we mean the theoretical possibility that complex cases might be pushed to the following year if the QBP funds were exhausted, or, conversely, less complex electives cases might be preferentially selected. This is an interesting question for future research, but beyond the scope of this paper. 

Q 8c: “Is the remuneration per patient adjusted for patient characteristics or the complexity of the condition?”

A 8c: There is a per hospital adjustment to account for overall complexity of the overall patient population, but there is no adjustment per patient. The QBP calculation is adjusted for each hospital, but not for each patient. CMI of the hospital (a hospital-level attribute based on historical patient case mix for similar admissions) is considered in the QBP fee. The amount of funding for the same QBP varies significantly from hospital to hospital based on CMI. We have added a sentence to clarify that, “The fee is adjusted for each hospital based on its case-mix index (CMI) to account for overall complexity in the patient population.”

9: It would also be helpful to understand the ability of hospital management to direct the clinical care of patients in their hospitals. How much discretion did they have?

Authors’ Response 9:

Hospital management has no discretion to direct the care of individual patients in their hospitals, but they have some discretion over, for example, discharge policies to encourage shorter lengths of stay, including resources on the ward to facilitate discharge. They also have some control over the OR time available. Again, the extent to which managerial discretion may influence clinical care is very interesting, but beyond the scope of this paper. 

10: Information on the nature of the intervention and how it would have affected clinical decision-making in the hospital might go some ways to explaining the results.

Authors’ Response 10:

With regard to the nature of the intervention, we have discussed this extensively in our prior work noted are references 7 and 12. We did not study clinical decision-making in hospital, per se, to understand the extent to which this might explain our results. This would require qualitative research, interviewing clinicians to understand how QBPs influenced their decision-making, if at all. This is beyond the scope of this particular paper. 

11: Second, before and after designs, such as the kind used in this study, face the challenge of distinguishing the effect of the intervention from the effects of other interventions that were introduced over the sample period. The authors describe some of these other interventions but are of the opinion that they can be safely ignored because i) none were introduced at the exact same time as the QBP policy and ii) those that were introduced before the policy would have effects that “would have been captured in the secular trend” and iii) those that were introduced after the policy would have effects that “did not affect the entire system in a reliable fashion for the procedures, diagnoses, or outcomes under investigation, or did not have a specific time when changes in the outcomes measured might have been expected.”

I don’t find these arguments persuasive. There are multiple ways that the QBP effect could be obscured by other effects. There is no requirement for instance that any policies introduced prior to the intro of the QBP would have effects that would combine and result in a continuous pre policy linear time trend. The statement that policies introduced after QBP had no impact on trends because one cannot predict when changes in outcomes would have occurred suggests that indeed these other policies could have had some impact.

I would address this issue in two ways. First, provide some more info on the other policies that were introduced over the sample period and provide some justification that they indeed have no material impact on the trends in the outcomes under investigation. This could be contained in an appendix.

Authors’ Response 11:

We can think of 3 possible options to address Reviewer #1’s request for more information on the other policies and further justification for why we believe they had no material impact. Our preference would be Option 1, but we defer to the editor to advise us on which of these is preferred:

Option 1: No further change required 

In Limitations, we already identify the initiatives that occurred near the implementation of QBPs, each with corresponding citations for readers interested in deeper examination. We accept that the reviewer is not convinced, but we have now re-examined each initiative, and we remain confident in what we already say in the manuscript. That is, “…we did not consider them to be potential temporal confounders nor a source of bias or threat to the internal validity of our ITS. We did not, therefore, evaluate any influence they may have had on our outcomes during the time of the funding change.” 

Importantly, the nature of these initiatives is such that they were mainly high level, implemented over a long time period – or not even fully implemented at all – across the entire system, and were very much indirectly related – if at all – to the QBPs and outcomes we evaluated. We did not evaluate the extent to which, or when, each of these initiative was implemented across the system, as that would have required an entirely different study. 

For example, the 2010 Excellent Care for All Act (enacted 2 years prior to implementation of the first QBPs), simply sets out a number of requirements for hospitals, including that they establish committees on quality-related issues, implement quality improvement plans, link executive compensation to achievement of targets, implement of patient satisfaction and staff surveys, declare their values following public consultation, and establish a patient relations process. We do not believe that these high level and long term efforts to put “patients first” and strengthen organization focus on accountability would have affected the four QBPs we studied, or the specific clinical outcomes we measured. 

Similarly, Health Links, introduced in 2014 (2 years after QBPs were implemented), focuses on coordinating care for patients living with multiple chronic conditions and complex needs, by ensuring that each patient has a Coordinated Care Plan (CCP). Again, it is unlikely that implementing a CCP would obscure any effects of the QBPs we studied (congestive heart failure, hip fracture, pneumonia, and prostate cancer surgery), for the outcomes we measured (return to hospital or death within 30 days, acute LOS, volume of admissions, and patient characteristics).

Similarly, pressure for supply side controls existed during a period of austerity that began in 2009, and hospitals contributed to getting the province back on track financially by accepting “years of zero percent funding increases at a time when inflation, patient volumes, labour costs, energy, and regulatory requirements grew significantly” (ref 36 in manuscript). Again, we don’t believe these economic ups and downs, pervasive in these times, would have obscured the effect for the QBPs and outcomes we studied. We have modified one sentence in the Limitations to clarify one of the initiatives, further explaining that there was “pressure to implement” supply side controls on hospital expenditures “during a period of austerity” from “2009” (not 2011, per reference 36). 

Similarly, the Health Based Allocation Methods (HBAM), introduced in 2012, was another new way to partially fund hospitals. Instead of just counting each patient that a hospital treats, HBAM weights cases to account for the fact that some hospitals treat more high-resource patients than others. HBAM uses data that trails funding by two years. Therefore, services changes in fiscal year 2012/13 would not impact funding until 2014/15. Our study period ran from 2010-2017. Any potential influence of HBAM could not even have occurred until some indeterminate time after 2014, and even then, HBAM and QBPs are completely separate funding streams that don't depend on, or compete with, each other. 

We do already say in the manuscript that, “However, these initiatives were mostly implemented well before the QBP funding reform, meaning that in this study their effects would have been captured in the secular trend. Those that occurred afterwards did not affect the entire system in a reliable fashion for the procedures, diagnoses, or outcomes under investigation, or did not have a specific time when changes in the outcomes measured might have been expected.”

Option 2: Add more to Limitations

We could add more details in Limitations including more citations and explanation about why these other high level policies were unlikely to have had any material impact on the trends we investigated. For example, similar to the explanations of these initiatives that we have provided in Option 1 above, we could say more about why we believe that any effects of policies introduced prior to QBPs would combine and result in a continuous pre-policy linear time trend. In this case, we seek advice from the Editor about what additional information is preferred, given that we do already provide citations to each of these initiatives for curious readers. 

Option 3: Add Appendix 

In an Appendix, we could separately describe in detail each of the other initiatives that were implemented during the study period, similar to what we have written in Option 1 above. However, we already cite source materials in the manuscript for each initiative, and readers could simply review the cited materials as is typical when a reader is curious to know more. 

12. Second it may be helpful as well to run a structural break test on the time series data, allowing the data to determine if there were any structural breaks at any point in the time series data. The references contained in the following study may be helpful

https://link.springer.com/article/10.1007/s10614-011-9271-1

In the cases where the authors have detected a break in trend (such as for hip fracture mortality), it would be useful to see if the automated structural break detection algorithm also detects this. This would lend some confidence that the break really is due to the QBP policy.

Authors’ Response 12: Change-point detection methods in time series are a popular method for detecting if any abrupt changes occurred at all during any time period. However, given the number of outcomes and QBPs we were studying, and the slow/uncertain implementation of any other changes during the study period (as explained above in Response 11, Option 1) we chose to pre-specify a time period, and incorporated an implementation period to detect the magnitude of change at the same time the policy was implemented rather than taking a more data-driven approach. From the variety of outcomes and QBPs we chose, none had any major effects and we cautiously interpret this as QBP having small, if any, effects.

13. The referee form also requires that I comment on the steps taken by the researchers to make their data available to other researchers so as to allow for independent replication. The authors have elected to keep their “source data” – both the individual level data and the aggregated data – confidential. They do display graphs of the trends of the deseasonalized data. But do not present the actual values of the aggregated data in either raw or deseasonalized forms. The claimed justification for the secrecy is a provision of Ontario’s “Personal Health Information Privacy Act” which evidently prohibits disclosure. My sense is that the study authors do not justify why disclosure is not permitted. I am very interested to understand exactly how the disclosure of data with personal identifiers removed could violate the Act. Exactly what is the privacy breach for releasing the individual level data with personal identifiers removed? How would the Act be violated if the aggregated unadjusted data were to be released?

Authors’ Response 13: We consulted with ICES’ Privacy Office and the Chief Science Officer. Our inability to make the aggregate unadjusted data sets publicly available has nothing to do with Ontario’s privacy laws, per se. Rather, it is currently prohibited under our data sharing agreements with data partners. While we renegotiate data sharing agreements to make more open access possible, access to study-specific data sets can be made available to individuals who meet criteria for confidential access through ICES’ Data and Analytical Services Unit. We described ICES policy on data sharing above, and repeat it here:

The dataset from this study is held securely in coded form at ICES. While data sharing agreements prohibit ICES from making the dataset publicly available, access may be granted to those who meet pre-specified criteria for confidential access. 

For more details, please also see: https://www.ices.on.ca/DAS/Public-Sector

Reviewer #2: General Comments:

14. QBP's and quality measures/clinical handbooks. The content of the QBP quality handbooks will not be familiar to many readers. Can the authors provide basic information on the content, scope, quality/rigor, and specificity of the clinical guidance in the QBP quality handbooks, perhaps in an online supplement? To what extent would the quality of care recommendations differ from the standard of care already in place in Canadian teaching hospitals or large community hospitals? How evidence-based and specific are the recommendations? Was there ever a realistic prospect that providing information in QBP handbooks could measurably improve quality of care, with or without an enforcement mechanism in QBP administration?

Authors’ Response 14:

In the manuscript, we already cite each of the four relevant QBP handbooks at references 18-21. We have now cited the handbooks a second time, at their first mentioned in the Introduction. 

The handbooks extensively describe the methodology by which they were created, including “content, scope, quality/rigor, and specificity of the clinical guidance”. We have added new text providing basic information about the handbooks, along with a new citation, as follows: “To create these handbooks for each QBP, the Ministry of Health and Long Term Care, in collaboration with partners such as Health Quality Ontario, Cancer Care Ontario, and the Cardiac Care Network, established expert advisory panels with leading clinicians, scientists, and patients. They defined episodes of care for selected diagnoses or procedures, developed best practice recommendations for patient care, and suggested indicators to monitor for ongoing quality improvement. The resulting QBP Clinical Handbooks serve as a compendium of evidence and clinical consensus.” 

We did not assess the extent to which the handbook recommendations differed from existing standards of care in place in Canadian teaching/community hospitals. This is a different study question, beyond the scope of ours.

In our previously published qualitative work we examined whether “there was ever a realist prospect that the QBP handbooks could measurably improve quality of care, with or without an enforcement mechanism”. We have now added a citation to this work as indicated in the Introduction. 

15. Is there a reason why QBP implementation would be expected to affect the number of admissions for congestive heart failure, pneumonia, hip fracture, and even prostatectomy? I would think that the number of admissions would be driven primarily by the disease incidence, rather that hospital capacity or funding models. More elective procedures, such as non-cancer surgery e.g. knee and hip joint replacement, might be more sensitive to changes in funding models.

Authors’ Response 15:

QBPs were originally selected in an attempt to standardize care for costly high volume diagnoses and procedures. Recall that the goals of QBPs were two-fold, as we explain in the Introduction: improve quality and reduce cost. We examined the tension between standardizing costs vs. standardizing quality in both of our published qualitative research papers (https://health-policy-systems.biomedcentral.com/articles/10.1186/s12961-018-0353-6 and https://journals.plos.org/plosone/article?id=10.1371/journal.pone.0191996), which we reference in this manuscript. 

We selected a range of QBP diagnoses and procedures – some planned, some unplanned, some surgical, some medical – in an attempt to assess whether the effects varied with the nature of the QBP. We assessed volume of admissions because we were interested in knowing whether QBPs might incentivize more admissions for some procedures and less for others, and whether that affected equitable access to care via adverse risk selection for patients > 65 years old, or those in lower income quintiles, or those with greater severity of illness. As we say in our Conclusion, going forward, it might be more useful to focus on “specific diagnoses and procedures”. This is partly what our research revealed. 

16. While implementation of QBPs did not affect the outcomes the authors evaluated, it is possible that the funding model did achieve other policy objectives. To the extent that QBPs are intended to limit the financial risk of the provincial funder associated with shortfalls in annual hospital budgets, it is possible that QBP funding did achieve some policy objectives. Further, it is possible that episode-of-care funding incentivized hospitals to provide more efficient and less costly care for these episodes. The investigators would not have detected these efficiencies, since they would not likely be reflected in changes in clinical outcome, or even general resource utilization measures such as length of stay.

Authors’ Response 16:

We agree. There may be other benefits or harms of QBPs that we did not measure, or other policy objectives that may have been met, such as related to total cost per episode-of-care or to the system overall. As we state in the first sentence of the “Summary of Findings”, we were careful to limit our Conclusions to only the QBPs and outcomes we evaluated to avoid being overbroad in our conclusions. 

Specific Comments:

17. Abstract "Patients from the lowest income neighborhoods increased slightly..." will be a confusing statement to people not familiar with SES income quintiles. Perhaps state something like "The proportion of patients admitted for congestive heart failure who lived in the lowest neighborhood income quintile increased.." or something like that which more clearly indicates the direction of any change in access/SES of target population.

Authors’ Response 17:

We have revised accordingly. The sentence now reads, “The percentage of patients who lived in the lowest neighborhood income quintile increased slightly for those diagnosed with congestive heart failure (1.89%; 95% CI: 0.51% to 3.27%) and decreased for those who underwent prostate cancer surgery (-2.08%; 95% CI: -3.74% to -0.43%).”

18. Introduction 3rd paragraph "Funding was carved out of hospitals' global budgets and then reallocated…as a fixed fee" is not entirely correct, since funding per episode of care is a product of a fixed fee and the CMI of the hospital (a hospital-level attribute based on historical patient case mix for similar admissions). The amount of funding for the same QBP varies significantly from hospital to hospital based on CMI.

Authors’ Response 18:

We have revised the Introduction as follows, “The fee is adjusted for each hospital based on its unique case-mix index (CMI) to account for the complexity in its overall patient population.” 

19. Page 7 What is "responsiveness" to QBP's? ("prior qualitative work which identified sources of potential variation in responsiveness to QBP's")

Authors’ Response 19:

We have revised the Methods as follows, “This was further informed by our prior qualitative work which identified sources of potential variation in the extent to which, and the ways in which, hospitals responded to QBPs.”

20. Page 9 explanation of access to care measures. More explanation about why the "proportion of patients living in lowest neighborhood income quintile" is a measure of access would help average readers understand this concept.

Authors’ Response 20: 

We have revised the Methods as follows:

“Socioeconomic status (SES) may contribute to inequalities in access to care, so we used neighborhood income quintile is an indicator of SES”.

21. Page 9 coding behaviour. Can you explain why mean HBAM HIG RIW trend is a measure of coding "behaviour"? How can you know if changes in RIW's do not represent true abrupt changes in patient case mix?

Authors’ Response 21:

As we say in Methods, “…to the extent changes in HIG weight do not represent true abrupt changes in patient case mix, the weight is one potential measure by which to evaluate effects on coding.”

HIG weight is a measure of coding “behaviour” because if upcoding is occurring—whether due to legitimately better coding or questionable or inappropriate coding—we would expect to see a change in HIG weight, since HIG weights are based on CMGs (which incorporate case mix), and upcoding would be reflected in case mix.

RIW is a “relative cost weight value assigned to each patient care episode. It reflects the resource intensity of each patient care episode and is adjusted for a number of factors (including age, comorbidity level and selected interventions).” (https://www.cihi.ca/en/pce_methodology_notes_en.pdf)

Although we observed no statistically significant changes in mean HIG weight for any of the four cohorts, had we seen changes in HIG weight, we know they would not have represented true abrupt changes in patient case mix because there were no changes in the patient characteristics that are used to calculate RIW (i.e. age, Charlson Comorbidity Index). Therefore, there was no abrupt change in patient case mix that would have accounted for changes in HIG weight, had we seen such changes (which we did not). In other words, if changes in HIG were associated with changes in case mix, it is likely we would have seen some evidence of this, as we considered trends in important patient characteristics. We saw neither.

upcoding

22. Page 11 results 2nd paragraph "...the effect of QBPs on outcomes 2 years after implementation..." (don't need to say "after 2 years post-implementation"). Same comment applies throughout eg 1st sentence 3rd paragraph of results.

Authors’ Response 22:

Prior to submitting the manuscript, we had considerable discussion about this language, including the precise meaning of “at” vs “after”, and the fact that the 25th data point was in April of the relevant year. We now suggest “at 2 years post-implementation”, and have revised accordingly. 

23. Page 14 discussion. "The slight variation we observed in effects--some results went up, others down...". These changes may still reflect noise rather than signal. I would not necessarily assume any causal association with the exposure. Other explanations do include random chance, bias, other secular trends, etc.

Authors’ Response 23: 

We agree and have added, “They may also represent noise rather than signal”. 

24. Page 15 "...the goal of the reform was to effect broad change..." (not affect)

Authors’ Response 24: 

We agree, and have revised accordingly.

---

## [Decision Letter · Decision Letter 1]

19 Feb 2020

PONE-D-19-27876R1

Effects of Quality-Based Procedure Hospital Funding Reform in Ontario, Canada: An Interrupted Time Series Study

PLOS ONE

Dear Dr Li,

Thank you for submitting your manuscript to PLOS ONE. After careful consideration, we feel that it has merit but does not fully meet PLOS ONE’s publication criteria as it currently stands. Therefore, we invite you to submit a revised version of the manuscript that addresses the points raised during the review process.

One of the reviewer has raised major concerns about the manuscript and particularly that his/her suggestions have not been appropriately addressed. You suggested different options how to proceed and the option of not including the suggestions is not appropriate. I, therefore, would like to ask you to reconsider your revisions. In its present form, it is not acceptable for publication.

We would appreciate receiving your revised manuscript by Apr 04 2020 11:59PM. To enhance the reproducibility of your results, we recommend that if applicable you deposit your laboratory protocols in protocols.io, where a protocol can be assigned its own identifier (DOI) such that it can be cited independently in the future. For instructions see: http://journals.plos.org/plosone/s/submission-guidelines#loc-laboratory-protocols

We look forward to receiving your revised manuscript.

Kind regards,

Hans-Peter Brunner-La Rocca, M.D.

Academic Editor

PLOS ONE

Reviewers' comments:

Reviewer's Responses to Questions

**Comments to the Author**

1. If the authors have adequately addressed your comments raised in a previous round of review and you feel that this manuscript is now acceptable for publication, you may indicate that here to bypass the “Comments to the Author” section, enter your conflict of interest statement in the “Confidential to Editor” section, and submit your "Accept" recommendation.

Reviewer #1: (No Response)

Reviewer #2: All comments have been addressed

2. Is the manuscript technically sound, and do the data support the conclusions?

Reviewer #1: Partly

Reviewer #2: Yes

3. Has the statistical analysis been performed appropriately and rigorously? 

Reviewer #1: I Don't Know

Reviewer #2: Yes

4. Have the authors made all data underlying the findings in their manuscript fully available?

Reviewer #1: No

Reviewer #2: (No Response)

5. Is the manuscript presented in an intelligible fashion and written in standard English?

Reviewer #1: Yes

Reviewer #2: (No Response)

6. Review Comments to the Author

Reviewer #1: The initial version of the paper provided little information on the nature of the QBP policies and the impacts of these policies on how patients would be cared for and managed or the types of patients that are admitted to hospital. Thus I asked the authors to expand their discussion of the impact of the QBP policies on the hospitals finances – which would be of obvious concern to the hospital management – and also discuss the influence, if any, hospital management had on the clinical decision making of healthcare providers working in the hospital. Obviously if the QBP policy has no impact on how patients are managed and treated in hospital then one does not need to conduct any empirical analysis. The policy, by design, will have no impact.

The authors seem to dismiss these concerns. When asked to provide more information on the nature of the financial arrangements between the government and hospitals the authors respond: “Although Reviewer #1 asks good questions, we feel that revising the manuscript to drill down to this level of detail about the flow of funds back and forth between government and hospitals goes beyond what is required in this manuscript on quantitative effects.”

When asked to provide more information on the control that hospital management has on healthcare providers, the authors respond: “... the extent to which managerial discretion may influence clinical care is very interesting, but beyond the scope of this paper.”

I disagree. To my mind, these are very much within the scope of the paper.

The initial version of the paper provided what I deemed to be an unconvincing explanation that the QBP effect was not confounded by other changes to hospital financing in the province of Ontario. To be clear, I am not suggesting here that the QBP effect was confounded. I was merely asking that the authors enumerate the other major policy changes that occurred over the sample period and provide some assurance that the effects of these policies, if any, could be relegated to the pre-policy linear time trend. This material could appear in an appendix.

The authors again elected to not make the change. They instead appealed to the journal editor to allow them to choose “Option 1: No further change required”. They are of the opinion that it is sufficient to state that “…we did not consider them to be potential temporal confounders nor a source of bias or threat to the internal validity of our ITS.” The authors take the view that the onus is on the reader to track down source material, contained in their reference list, and then make an independent determination. I disagree. My view is that the onus is on the authors to provide some evidence that the policy effects are identified. The authors have gone some way in the direction that I recommended. In their reply letter, the authors have enumerated each of the other policies and provided some discussion. This could easily be expanded on to form materials for a supplementary appendix.

My final major comment on the initial version of the paper concerned the inability of the reader to access even the highly aggregated data for independent replication of their results and the adequacy of their regression model specification. The authors do graph the deseasonalized data but the data points are faint, making it hard to check their model fit and specification. The authors also refused my request to conduct tests for structural breaks at other time points.

Part of the stated rationale for keeping even the highly aggregated data secret was that Ontario’s privacy laws precluded their disclosure. I challenged the authors on this. The authors have now removed this rationale from the paper and now stress that agreements with “data partners” are the limiting factor. This explanation, too, seems questionable. What aspect of the data sharing agreements permit exposition of graphs of the deseasonalized time series data (albeit graphs rendered in a way that makes it difficult to ascertain the data values) but prohibit actually providing the unadjusted aggregated data?

In summary, then, the authors, in the revised paper, have rejected, without sufficient justification, my primary suggestions:

• exposition of discussion of how the QBP would be expected to affect patient health outcomes due to their impact on hospital finances and the attendant impact of these changes in managerial incentives on clinical care

• exposition of the role of concurrent policies to obscure the apparent impact of QBP on outcomes

• evaluation of their ITS model by allowing for different break points in the time series

• disclosure of even the aggregated de-seasonalized time series data to permit independent verification

Reviewer #2: The manuscript contains changes responsive to the reviewer comments. I do note that the authors have provided answers to several author queries in their response to reviewers but have not included all of thin information in the manuscript. The authors should consider including information summarizing their responses in the revised manuscript, where appropriate. (Readers are likely to have similar questions as the manuscript reviewers)

7. PLOS authors have the option to publish the peer review history of their article (what does this mean?). If published, this will include your full peer review and any attached files.

Reviewer #1: No

Reviewer #2: Yes: David Urbach

---

## [Author Response · Author response to Decision Letter 1]

5 Apr 2020

Dear Dr. Brunner-La Rocca, 

Thank you for the opportunity to revise and resubmit our manuscript. 

Please find attached our reply, which we hope you will find satisfactory.

Best wishes,

Alvin Li, Karen Palmer, and Noah Ivers 

 

Authors’ response to Reviewers

Reviewer #1: 

The initial version of the paper provided little information on the nature of the QBP policies and the impacts of these policies on how patients would be cared for and managed or the types of patients that are admitted to hospital. Thus I asked the authors to expand their discussion of the impact of the QBP policies on the hospitals finances – which would be of obvious concern to the hospital management – and also discuss the influence, if any, hospital management had on the clinical decision making of healthcare providers working in the hospital. Obviously if the QBP policy has no impact on how patients are managed and treated in hospital then one does not need to conduct any empirical analysis. The policy, by design, will have no impact.

The authors seem to dismiss these concerns. When asked to provide more information on the nature of the financial arrangements between the government and hospitals the authors respond: “Although Reviewer #1 asks good questions, we feel that revising the manuscript to drill down to this level of detail about the flow of funds back and forth between government and hospitals goes beyond what is required in this manuscript on quantitative effects.”

When asked to provide more information on the control that hospital management has on healthcare providers, the authors respond: “... the extent to which managerial discretion may influence clinical care is very interesting, but beyond the scope of this paper.”

I disagree. To my mind, these are very much within the scope of the paper.

The initial version of the paper provided what I deemed to be an unconvincing explanation that the QBP effect was not confounded by other changes to hospital financing in the province of Ontario. To be clear, I am not suggesting here that the QBP effect was confounded. I was merely asking that the authors enumerate the other major policy changes that occurred over the sample period and provide some assurance that the effects of these policies, if any, could be relegated to the pre-policy linear time trend. This material could appear in an appendix.

The authors again elected to not make the change. They instead appealed to the journal editor to allow them to choose “Option 1: No further change required”. They are of the opinion that it is sufficient to state that “…we did not consider them to be potential temporal confounders nor a source of bias or threat to the internal validity of our ITS.” The authors take the view that the onus is on the reader to track down source material, contained in their reference list, and then make an independent determination. I disagree. My view is that the onus is on the authors to provide some evidence that the policy effects are identified. The authors have gone some way in the direction that I recommended. In their reply letter, the authors have enumerated each of the other policies and provided some discussion. This could easily be expanded on to form materials for a supplementary appendix.

My final major comment on the initial version of the paper concerned the inability of the reader to access even the highly aggregated data for independent replication of their results and the adequacy of their regression model specification. The authors do graph the deseasonalized data but the data points are faint, making it hard to check their model fit and specification. The authors also refused my request to conduct tests for structural breaks at other time points.

Part of the stated rationale for keeping even the highly aggregated data secret was that Ontario’s privacy laws precluded their disclosure. I challenged the authors on this. The authors have now removed this rationale from the paper and now stress that agreements with “data partners” are the limiting factor. This explanation, too, seems questionable. What aspect of the data sharing agreements permit exposition of graphs of the deseasonalized time series data (albeit graphs rendered in a way that makes it difficult to ascertain the data values) but prohibit actually providing the unadjusted aggregated data?

In summary, then, the authors, in the revised paper, have rejected, without sufficient justification, my primary suggestions:

Point #1. exposition of discussion of how the QBP would be expected to affect patient health outcomes due to their impact on hospital finances and the attendant impact of these changes in managerial incentives on clinical care

Point #2. exposition of the role of concurrent policies to obscure the apparent impact of QBP on outcomes

Point #3. evaluation of their ITS model by allowing for different break points in the time series

Point #4. disclosure of even the aggregated de-seasonalized time series data to permit independent verification

Response to Reviewer #1: 

We offer the following responses to each of the Reviewer’s four primary suggestions: 

Point #1. With regard to “exposition of discussion of how the QBP would be expected to affect patient health outcomes due to their impact on hospital finances and the attendant impact of these changes in managerial incentives on clinical care:

The Ontario Ministry of Health and Long Term Care (MoHLTC) hypothesized that by providing a standard price per diagnosis/procedure, along with a handbook of best practice clinical care pathways to guide clinical care and reduce variation in care, the types of outcomes we measured would improve. The point of QBPs was simply to set the fee paid per diagnosis/procedure (because there was presumed to be too much variation in cost) and incentivize best practice (because the variation in cost was presumed to be associated with too much variation in care). The MoHLTC’s rationale and assumed mechanism of action for QBPs was as follows:

"QBPs are specific clusters of patient services that offer opportunities for health care providers to share best practices and will allow the system to provide even better quality care, while increasing system efficiencies. By promoting the adoption of clinical evidence-informed practices, clinical practice variation should be reduced across the province while improving patient outcomes to ensure that patients receive the right care, in the right place, at the right time."At page 4, Introduction, we have added this quote to the manuscript to further explain QBPs.

The broader impetus for Health System Funding Reform, of which QBPs were a component, was: 

“to promote quality and improved outcomes and create a more equitable allocation of resources. Many countries around the world, including Australia, Germany, Denmark and the United Kingdom have used funding as a lever for change. Over the past two decades, these models have been associated with successes in decreasing wait times/ improving access to care, reducing unit costs per admission, reducing variation in both costs and clinical practice and, most importantly, improving quality.”

Our intent was to test the MoHLTC’s hypothesis that QBPs would facilitate best practice, which they believed would improve these patient outcomes. To date, there has been no peer-reviewed evaluation of the overall effects of QBPs on key indicators of patient care. We simply took advantage of Ontario’s data infrastructure to evaluate whether this new hospital payment model aimed at system-level change had any impact on measures of quality of care, access to care, and hospital coding behaviour for four QBPs.

The Reviewer asks that we explain the “impact of the QBP policies on the hospitals’ finances”. We did not assess how QBPs impacted hospitals’ finances/budgets. We don't know whether or how QBPs changed hospitals’ budgets, so we cannot make any inferences about whether increases or decreases in hospitals’ budgets were associated with improvement or worsening of the indicators we evaluated. Similarly, since we did not assess the impact of QBPs on hospitals’ finances, we do not known whether there were budget-induced changes in managerial incentives that might have affected clinical care. To further answer the reviewer’s question about the “nature” of those arrangements would require qualitative interviews with government and hospitals’ CFOs. Lastly, we didn’t study the extent to which hospital management controls, or does not control, health care providers in hospitals. To assess this would require qualitative interviews with hospital clinicians and hospital management. In general, though, in Canadian hospitals physician clinicians function with autonomy in their decision-making about patient care.

At Page 6, Limitations, we have now added a sixth point to explain this:

“Sixth, we did not assess how QBPs impacted hospitals’ finances, so we cannot make any inferences about whether increases or decreases in hospitals’ budgets affected patient care and/or outcomes for the QBPs we evaluated.”

Point #2: With regard to “exposition of the role of concurrent policies to obscure the apparent impact of QBP on outcomes”:

We have added a Supplemental Table with an overview of concurrent policies and now reference this table in the Limitations section. 

S7 Table: Overlapping Policies

Concurrent Initiative Description

Excellent Care for All Act (2010) This initiative was an important policy change, which was passed on 2010. 

“The Act requires health care organizations, currently defined as hospitals, to:

• Develop and post annual quality improvement plans.

• Implement patient and employee satisfaction surveys and a patient relations process.

• Link executive compensation to achievement of quality plan performance improvement targets.

• Develop declarations of values after public consultation.

• Create quality committees to report to each hospital board on quality related issues.”

Source: https://www.ontariocanada.com/registry/view.do?postingId=4544&language=en

Community Health Links, (2014) This program provides individualized, coordinated, care plans for patients living with multiple chronic conditions and complex needs. 

Source: http://www.health.gov.on.ca/en/pro/programs/transformation/community.aspx

Health Based Allocation Methods

(2012) Health Based Allocation Methods (HBAM) was a new funding methodology to partially fund hospitals. HBAM uses expected weighted cases and expected unit cost. 

Source: https://www.oha.com/Documents/HBAM-What%20You%20Need%20To%20Know.pdf

Austerity measures (since 2009) “During a period of significant austerity beginning in 2009, Ontario’s hospitals contributed to getting the province back on track financially by accepting years of zero percent funding increases at a time when inflation, patient volumes, labour costs, energy, and regulatory requirements grew significantly” 

More info: https://www.oha.com/Bulletins/2558_OHA_A%20Sector%20on%20the%20Brink_rev.pdf

Point #3: With regard to “evaluation of their ITS model by allowing for different break points in the time series”:

At page 6, Methods, we have revised to justify our rationale for using a quasi- experimental ITS approach, as follows: 

 “Rather than a data driven approach, we chose to pre-specify time periods simultaneous with when each QBP was introduced. A priori, we decided to incorporate a 3 month transition period to allow time for any clinical changes in response to the funding model change to be implemented. Had we used a data-driven approach, it is likely that, given the number of outcomes and QBPs, we would have detected changes during the time series and would not have been able to infer that these changes were a result of effects from QBPs. Instead, we used an ITS design because, in the absence of randomized experiments, ITS design is an effective method to evaluate policy changes at the whole system- and population-level.[18–22].” 

Point #4: With regard to “disclosure of even the aggregated de-seasonalized time series data to permit independent verification”:

Below we now include a letter from the ICES’ Privacy & Legal Office that explains the basis for ICES’ restriction on release of datasets for public use. PlosOne has challenged ICES’ justification in the past, and we understand that much time has been consumed by ICES and other authors trying to explain the nuances to different personnel. Thus, we have also sent this letter to the Editor-in-Chief, Dr. Joerg Heber for consideration (with copy to Kelley Ross, ICES’ Privacy & Legal Office). We understand the frustration of not disclosing even the aggregated de-seasonalized data, given the ongoing open science initiative, which we support wholeheartedly. On this matter, however, we, as scientists, have no choice. 

Reviewer #2: 

The manuscript contains changes responsive to the reviewer comments. I do note that the authors have provided answers to several author queries in their response to reviewers but have not included all of thin information in the manuscript. The authors should consider including information summarizing their responses in the revised manuscript, where appropriate. (Readers are likely to have similar questions as the manuscript reviewers)

Response to Reviewer #2: Thank you for your comment. We now added selected information from our previous responses into the revised manuscript, as indicated in tracked changes and below: 

At page 9, Data Sources and Quality, we have further explained why HIG weight is a measure of upcoding, as follows:

“HIG weight is a measure of coding behaviour because it incorporates both case mix and the resource intensity of each patient care episode adjusted for patient characteristics. If upcoding is occurring we would expect to see changes in HIG weight. Thus, to the extent changes in HIG weight do not represent true abrupt changes in patient case mix, the HIG weight is one potential measure by which to evaluate effects on coding.”

At page 6, Limitations, we have added this:

“Seventh, there may be benefits or harms of QBPs that we did not measure, or other policy objectives that may have been met, such as those related to total cost per episode-of-care or cost to the system overall. We were careful to limit our conclusions to only the QBPs and outcomes we evaluated to avoid being overbroad.”

---

## [Decision Letter · Decision Letter 2]

30 Apr 2020

PONE-D-19-27876R2

Effects of Quality-Based Procedure Hospital Funding Reform in Ontario, Canada: An Interrupted Time Series Study

PLOS ONE

Dear Dr Li,

Thank you for submitting your manuscript to PLOS ONE. After careful consideration, we feel that it has merit but does not fully meet PLOS ONE’s publication criteria as it currently stands. Therefore, we invite you to submit a revised version of the manuscript that addresses the points raised during the review process.

One of the reviewers remain to have significant concerns about your manuscript. I would like you to adequately address these issues. Even if you disagree with the reviewer you must clearly state why the reviewer's notion are not correct and why you did not changed the revised manuscript accordingly.

We would appreciate receiving your revised manuscript by Jun 14 2020 11:59PM. To enhance the reproducibility of your results, we recommend that if applicable you deposit your laboratory protocols in protocols.io, where a protocol can be assigned its own identifier (DOI) such that it can be cited independently in the future. For instructions see: http://journals.plos.org/plosone/s/submission-guidelines#loc-laboratory-protocols

We look forward to receiving your revised manuscript.

Kind regards,

Hans-Peter Brunner-La Rocca, M.D.

Academic Editor

PLOS ONE

Reviewers' comments:

Reviewer's Responses to Questions

**Comments to the Author**

1. If the authors have adequately addressed your comments raised in a previous round of review and you feel that this manuscript is now acceptable for publication, you may indicate that here to bypass the “Comments to the Author” section, enter your conflict of interest statement in the “Confidential to Editor” section, and submit your "Accept" recommendation.

Reviewer #1: (No Response)

Reviewer #2: All comments have been addressed

2. Is the manuscript technically sound, and do the data support the conclusions?

Reviewer #1: Partly

Reviewer #2: Yes

3. Has the statistical analysis been performed appropriately and rigorously? 

Reviewer #1: No

Reviewer #2: No

4. Have the authors made all data underlying the findings in their manuscript fully available?

Reviewer #1: No

Reviewer #2: Yes

5. Is the manuscript presented in an intelligible fashion and written in standard English?

Reviewer #1: Yes

Reviewer #2: Yes

6. Review Comments to the Author

Reviewer #1: In this revised manuscript the authors write, in the methods section:

Had we used a data-driven approach, it is likely that, given the number of outcomes and QBPs, we would have detected changes during the time series and would not have been able to infer that these changes were a result of effects from QBPs. Instead, we used an ITS design because, in the absence of randomized experiments, ITS design is an effective method to evaluate policy changes at the whole system- and population-level.” 18–22]

I don't find this justification for not conducting the structural break tests convincing. The ITS design per se is not "an effective method to evaluate policy changes at the whole system- and population-level.” It depends critically on the extent to which the pre-policy trend, extrapolated into the post-policy period, reflect the counterfactual. It also depends on the pre policy trends being linear.

In the limitations section, the potential for confounding from the policy changes that occurred during the sample period. The authors write:

"However, these initiatives were mostly implemented well before the QBP funding reform, meaning that in this

study their effects would have been captured in the secular trend. Those that occurred afterwards

did not affect the entire system in a reliable fashion for the procedures, diagnoses, or outcomes

under investigation, or did not have a specific time when changes in the outcomes measured

might have been expected. Therefore, our findings regarding the (lack of) achievement of desired

system-level changes with the QBP funding reform in Ontario are robust."

As I have stated in my reviews of the earlier versions of the manuscript my view is that you can't draw this conclusion. It simply is not supported by the results that you present. You are unable or unwilling to render your graphs in a way that one can clearly see the data points or display the values of the aggregated data in the graphs. Your are unwilling to perform structural break tests even though you admit the possibility earlier on in the methods section that you might encounter breaks different from the ones that you pre-specify.

We do seem to be going in circles here. I would like to see these results in print but I am not satisfied with your identification strategy.

To break the impasse, I propose that you render the graphs so that the data points are more legible (this may involve shrinking the y scale and increasing marker size); the data points in the graphs in the paper I am reviewing are very faint. If you are able to do this then you can perform a visual inspection of the data points around the policy change. If the data points appear to be clustered around a linear trend over the sample period then this is good evidence that there were no effects from QBP. If there were discontinuities in the data series then you can comment on the role of QBP and the role of the other policies that were introduced around the same period of time and the other factors that could have affected outcomes (such as the role of provincial fiscal conditions, tax revenues and hospital budgets). If there is uncertainty over what caused a break in the time series after the intro of the QBP policies then you can estimate the size of the break and you can bound the size of the QBP effect. You have already stated that the QBP policy effect was not confounded, but if the QBP policy effect was realized only after a lag, and there were other factors changing around the same time, would these factors exert an increase or decrease on the outcome variable? What effect would these have on the apparent QBP policy effect?

Reviewer #2: (No Response)

7. PLOS authors have the option to publish the peer review history of their article (what does this mean?). If published, this will include your full peer review and any attached files.

Reviewer #1: No

Reviewer #2: Yes: David Urbach

---

## [Author Response · Author response to Decision Letter 2]

21 Jun 2020

Dear Dr. Brunner-La Rocca, 

Thank you for the opportunity to revise and resubmit our manuscript. 

Please find attached our reply. We’ve now addressed the reviewer’s concerns and include our analytic code and data creation plan as a supplemental information. We’ve also made a number of edits throughout the manuscript. We sincerely hope you will now find our manuscript acceptable for publication. 

Best wishes,

Alvin Li, Karen Palmer and Noah Ivers on behalf of co-authors

 

Reviewer #1: 

Reviewer Comment #1:

In this revised manuscript the authors write, in the methods section:

“Had we used a data-driven approach, it is likely that, given the number of outcomes and QBPs, we would have detected changes during the time series and would not have been able to infer that these changes were a result of effects from QBPs. Instead, we used an ITS design because, in the absence of randomized experiments, ITS design is an effective method to evaluate policy changes at the whole system- and population-level.” 18–22]

I don't find this justification for not conducting the structural break tests convincing. The ITS design per se is not "an effective method to evaluate policy changes at the whole system- and population-level.” It depends critically on the extent to which the pre-policy trend, extrapolated into the post-policy period, reflect the counterfactual. It also depends on the pre policy trends being linear.

RESPONSE #1: We have deleted that text from the revised version.

Reviewer Comment #2: In the limitations section, the potential for confounding from the policy changes that occurred during the sample period. The authors write:

"However, these initiatives were mostly implemented well before the QBP funding reform, meaning that in this study their effects would have been captured in the secular trend. Those that occurred afterwards did not affect the entire system in a reliable fashion for the procedures, diagnoses, or outcomes under investigation, or did not have a specific time when changes in the outcomes measured might have been expected. Therefore, our findings regarding the (lack of) achievement of desired system-level changes with the QBP funding reform in Ontario are robust."

As I have stated in my reviews of the earlier versions of the manuscript my view is that you can't draw this conclusion. It simply is not supported by the results that you present. 

RESPONSE #2: We have deleted this text from the revised version. 

Reviewer Comment #3: You are unable or unwilling to render your graphs in a way that one can clearly see the data points or display the values of the aggregated data in the graphs. Your are unwilling to perform structural break tests even though you admit the possibility earlier on in the methods section that you might encounter breaks different from the ones that you pre-specify.

We do seem to be going in circles here. I would like to see these results in print but I am not satisfied with your identification strategy.

To break the impasse, I propose that you render the graphs so that the data points are more legible (this may involve shrinking the y scale and increasing marker size); the data points in the graphs in the paper I am reviewing are very faint. If you are able to do this then you can perform a visual inspection of the data points around the policy change. 

If the data points appear to be clustered around a linear trend over the sample period then this is good evidence that there were no effects from QBP. If there were discontinuities in the data series then you can comment on the role of QBP and the role of the other policies that were introduced around the same period of time and the other factors that could have affected outcomes (such as the role of provincial fiscal conditions, tax revenues and hospital budgets). If there is uncertainty over what caused a break in the time series after the intro of the QBP policies then you can estimate the size of the break and you can bound the size of the QBP effect. You have already stated that the QBP policy effect was not confounded, but if the QBP policy effect was realized only after a lag, and there were other factors changing around the same time, would these factors exert an increase or decrease on the outcome variable? What effect would these have on the apparent QBP policy effect?”

RESPONSE #3: As the reviewer suggested, we have reproduced the graphs so that the data points are more visible. We have also carried out the requested visual inspection of the data points around the various policy changes and added additional ITS plots with vertical lines indicating the timing of these other policy changes to allow the reader to inspect the data. We have also made a number of cosmetic improvements to the figures. 

We see no obvious discontinuities associated with these other initiatives, with the possible exception of Health Links. We now state in the Limitations section: 

“Our study has several important limitations that are common to observational studies of policy changes. Teasing apart the effect of QBPs in the presence of multiple system-level changes is challenging. First, other initiatives to improve patient care and/or control costs may have overlapped with the timing of QBP implementation. Specific initiatives that we are aware of included passage in 2010 of the Excellent Care for All Act (ECFA)[40], the introduction of Health Based Allocation Model hospital funding reforms in April 2012 [24], and the introduction of Community Health Links in December 2012[42] (see S5-7 Figs). Visual inspection of data points around the timing of introduction of these initiatives however suggests that they are unlikely to have had a major impact on the outcomes studied in our analyses. A possible exception is the introduction of Community Health Links in December 2012: due to its timing close to that of the QBPs for congestive heart failure (CHF) in April 2013, it is difficult to independently assess the effect of the QBPs for this condition. Undetected confounding is always possible in any uncontrolled study. Policies aimed at improving health care are constantly being tinkered with, which may influence any particular intervention, such as QBPs, in ways not easily detected. 

We have included these new figures in the Supplemental Figures 5-7 and revised the Limitations text as above. 

The descriptions for the three figures are as follows and they are attached as a separate image file (per PLoS One policy).

S5 Fig. Percent of patients returned to hospital or died with Competing Initiatives. Red solid line represents the fitted model. The red dashed line represents the counterfactual (i.e. if no policy change occurred). The vertical dashed line represents the date of policy change. The grey shaded area represents the three months of “transition” period. Competing initiatives are outlined in the legend. Data are seasonally adjusted.

S6. Fig. Mean acute length of stay for the episode of care (days) with Competing Initiatives. Red solid line represents the fitted model. The red dashed line represents the counterfactual (i.e. if no policy change occurred). The vertical dashed line represents the date of policy change. Competing Interventions are outlined in the legend. The grey shaded area represents the three months of “transition” period. Competing Initiatives are outlined in the legend. Data are seasonally adjusted.

S7 Fig. Total volume of admissions with Competing Initiatives. Red solid line represents the fitted model. The red dashed line represents the counterfactual (i.e. if no policy change occurred). The vertical dashed line represents the date of policy change. The grey shaded area represents the three months of “transition” period. Competing Initiatives are outlined in the legend. Data are seasonally adjusted.

Data Sharing

We have now included the codes used in the analyses as a supplementary file.

---

## [Editor Report · Decision Letter 3]

9 Jul 2020

Effects of Quality-Based Procedure Hospital Funding Reform in Ontario, Canada: An Interrupted Time Series Study

PONE-D-19-27876R3

Dear Dr. Li,

We’re pleased to inform you that your manuscript has been judged scientifically suitable for publication and will be formally accepted for publication once it meets all outstanding technical requirements.

Kind regards,

Hans-Peter Brunner-La Rocca, M.D.

Academic Editor

PLOS ONE
---

## [Editor Report · Acceptance letter]

7 Aug 2020

PONE-D-19-27876R3 

Effects of Quality-Based Procedure Hospital Funding Reform in Ontario, Canada: An Interrupted Time Series Study 

Dear Dr. Li:

I'm pleased to inform you that your manuscript has been deemed suitable for publication in PLOS ONE. Congratulations! Your manuscript is now with our production department. 

Kind regards, 

on behalf of

Dr. Hans-Peter Brunner-La Rocca 

Academic Editor

PLOS ONE